# Sushi domain-containing protein 4 controls synaptic plasticity and motor learning

Inés González-Calvo[1,2†‡], Keerthana Iyer[1†], Mélanie Carquin[1], Anouar Khayachi[1], Fernando A Giuliani[2], Séverine M Sigoillot[1], Jean Vincent[3], Martial Séveno[4], Maxime Veleanu[1], Sylvana Tahraoui[1], Mélanie Albert[1], Oana Vigy[5], Célia Bosso-Lefèvre[1], Yann Nadjar[6], Andréa Dumoulin[6], Antoine Triller[6], Jean-Louis Bessereau[7], Laure Rondi-Reig[3], Philippe Isope[2], Fekrije Selimi[1*]

[1]Center for Interdisciplinary Research in Biology (CIRB), Collège de France, CNRS, INSERM, PSL Research University, Paris, France; [2]Institut des Neurosciences Cellulaires et Intégratives (INCI), CNRS, Université de Strasbourg, Strasbourg, France; [3]Institut Biology Paris Seine (IBPS), Neuroscience Paris Seine (NPS), CeZaMe, CNRS, Sorbonne University, INSERM, Paris, France; [4]BioCampus Montpellier, CNRS, INSERM, Université de Montpellier, Montpellier, France; [5]Institut de Génomique Fonctionnelle, CNRS, INSERM, Université de Montpellier, Montpellier, France; [6]École Normale Supérieure, Institut de Biologie de l'ENS, INSERM, CNRS, PSL Research University, Paris, France; [7]Université de Lyon, Université Claude Bernard Lyon 1, CNRS UMR 5310, INSERM U 1217, Institut Neuromyogène, Lyon, France

**\*For correspondence:**
fekrije.selimi@college-de-france.fr

[†]These authors contributed equally to this work

**Present address:** [‡]Univ. Bordeaux, CNRS, Interdisciplinary Institute for Neuroscience, IINS, UMR 5297, F-33000, Bordeaux, France

**Competing interests:** The authors declare that no competing interests exist.

**Abstract** Fine control of protein stoichiometry at synapses underlies brain function and plasticity. How proteostasis is controlled independently for each type of synaptic protein in a synapse-specific and activity-dependent manner remains unclear. Here, we show that *Susd4*, a gene coding for a complement-related transmembrane protein, is expressed by many neuronal populations starting at the time of synapse formation. Constitutive loss-of-function of *Susd4* in the mouse impairs motor coordination adaptation and learning, prevents long-term depression at cerebellar synapses, and leads to misregulation of activity-dependent AMPA receptor subunit GluA2 degradation. We identified several proteins with known roles in the regulation of AMPA receptor turnover, in particular ubiquitin ligases of the NEDD4 subfamily, as SUSD4 binding partners. Our findings shed light on the potential role of *SUSD4* mutations in neurodevelopmental diseases.

## Introduction

Proteostasis is at the core of many cellular processes and its dynamics need to be finely regulated for each protein in each organelle. In neurons, additional challenges are imposed by their spatial complexity. In particular, during long-term synaptic plasticity, the proposed substrate for learning and memory (*Collingridge et al., 2010*; *Nicoll, 2017*), the number of neurotransmitter receptors needs to be regulated independently in a synapse-specific and activity-dependent manner. At excitatory synapses, the modification of AMPA receptor numbers is a highly dynamic process, involving regulation of receptor diffusion (*Choquet and Triller, 2013*; *Penn et al., 2017*), their insertion in the plasma membrane, anchoring at the postsynaptic density and endocytosis (*Anggono and Huganir, 2012*). After activity-dependent endocytosis, AMPA receptors are either recycled to the plasma

membrane or targeted to the endolysosomal compartment for degradation (*Ehlers, 2000*; *Lee et al., 2004*; *Park et al., 2004*). The decision between these two fates, recycling or degradation, regulates the direction of synaptic plasticity. Recycling promotes long-term potentiation (LTP) and relies on many molecules such as GRASP1, GRIP1, PICK1, and NSF (*Anggono and Huganir, 2012*). Targeting to the endolysosomal compartment and degradation promote long-term depression (LTD; *Fernández-Monreal et al., 2012*; *Kim et al., 2017*; *Matsuda et al., 2013*), but the regulation of the targeting and degradation process remains poorly understood.

The Complement Control Protein (CCP) domain, an evolutionarily conserved module also known as Sushi domain, was first characterized in proteins with role in immunity, in particular in the complement system. In the past few years, proteins with CCP domains have been increasingly recognized for their role at neuronal synapses. Acetylcholine receptor clustering is regulated by CCP domain-containing proteins in *Caenorhabditis elegans* (*Gendrel et al., 2009*) and in *Drosophila melanogaster* (*Nakayama et al., 2016*). In humans, mutations in the CCP domain-containing secreted protein SRPX2 are associated with epilepsy and speech dysfunction, and SRPX2 knockdown leads to decreased synapse number and vocalization in mice (*Sia et al., 2013*). Recently SRPX2 has been involved in the regulation of synapse elimination in the visual and somatosensory systems (*Cong et al., 2020*). Despite the increase in the diversity of CCP domain-containing proteins in evolution (11 CCP domain-containing in *C. elegans* and 56 in humans; smart.embl.de), the function of many CCP domain-containing proteins remains unknown.

The mammalian *SUSD4* gene codes for a transmembrane protein with four extracellular CCP domains (*Figure 1A*) and is highly expressed in the central nervous system (*Holmquist et al., 2013*). The *SUSD4* gene is located in a genomic region deleted in patients with the 1q41q42 syndrome that includes developmental delays and intellectual deficiency (ID; *Rosenfeld et al., 2011*). *SUSD4* is also among the 124 genes enriched in de novo missense mutations in a large cohort of individuals with autism spectrum disorders (ASDs) or IDs (*Coe et al., 2019*). A copy number variation and several de novo mutations with a high CADD score, which indicates the deleteriousness of the mutations, have been described in the *SUSD4* gene in patients with ASDs (*Cuscó et al., 2009*; denovo-db, Seattle, WA [denovo-db.gs.washington.edu] 10, 2019). The SUSD4 protein has been described to regulate complement system activation in erythrocytes by binding the C1Q globular domain (*Holmquist et al., 2013*). Interestingly, this domain is found in major synaptic regulators such as C1QA (*Stevens et al., 2007*), CBLNs (*Matsuda et al., 2010*; *Uemura et al., 2010*), and C1Q-like proteins (*Bolliger et al., 2011*; *Kakegawa et al., 2015*; *Sigoillot et al., 2015*). Altogether these studies point to a potential role of SUSD4 in synapse formation and/or function and in the etiology of neurodevelopmental disorders.

Proper development and function of the cerebellar circuitry is central for motor coordination and adaptation, and a range of cognitive tasks (*Badura et al., 2018*; *Hirai et al., 2005*; *Ichise et al., 2000*; *Lefort et al., 2019*; *Rochefort et al., 2011*; *Tsai et al., 2012*). Cerebellar dysfunction is associated with several neurodevelopmental disorders including ASDs (*Stoodley, 2016*; *Stoodley et al., 2018*; *Wang et al., 2014*). In this circuit, cerebellar Purkinje cells (PCs) receive more than a hundred thousand parallel fiber (PF) synapses whose formation, maintenance, and plasticity are essential for cerebellar-dependent learning (*Gutierrez-Castellanos et al., 2017*; *Hirai et al., 2005*; *Ito, 2006*; *Kashiwabuchi et al., 1995*). Postsynaptic LTD was first described at synapses between PFs and cerebellar PCs (*Gao et al., 2012*; *Hirano, 2018*; *Ito, 2001*; *Ito and Kano, 1982*), where it can be induced by conjunctive stimulation of PFs with the other excitatory input received by PCs, the climbing fiber (CF; *Coesmans et al., 2004*; *Ito, 2001*; *Suvrathan et al., 2016*). The function of members of the C1Q family, such as CBLN1 and C1QL1, is essential for excitatory synapse formation and LTD in cerebellar PCs (*Hirai et al., 2005*; *Kakegawa et al., 2015*; *Matsuda et al., 2010*; *Sigoillot et al., 2015*; *Uemura et al., 2010*), suggesting that proteins such as SUSD4, that interact with the C1Q globular domain, could regulate these processes.

Gene expression studies from our laboratory revealed that *Susd4* is highly expressed in the olivo-cerebellar system of the mouse. In order to uncover the potential link between SUSD4 and neurodevelopmental disorders, we sought to identify the role of SUSD4 in brain development and function, by analyzing the phenotype of a *Susd4* constitutive loss-of-function mouse model. Here, we show that knockout (KO) of the *Susd4* gene leads to deficits in motor coordination adaptation and learning, misregulation of synaptic plasticity in cerebellar PCs, as well as an impairment in the degradation of GluA2 AMPA receptor subunits after chemical induction of LTD. Proteomic analysis of SUSD4

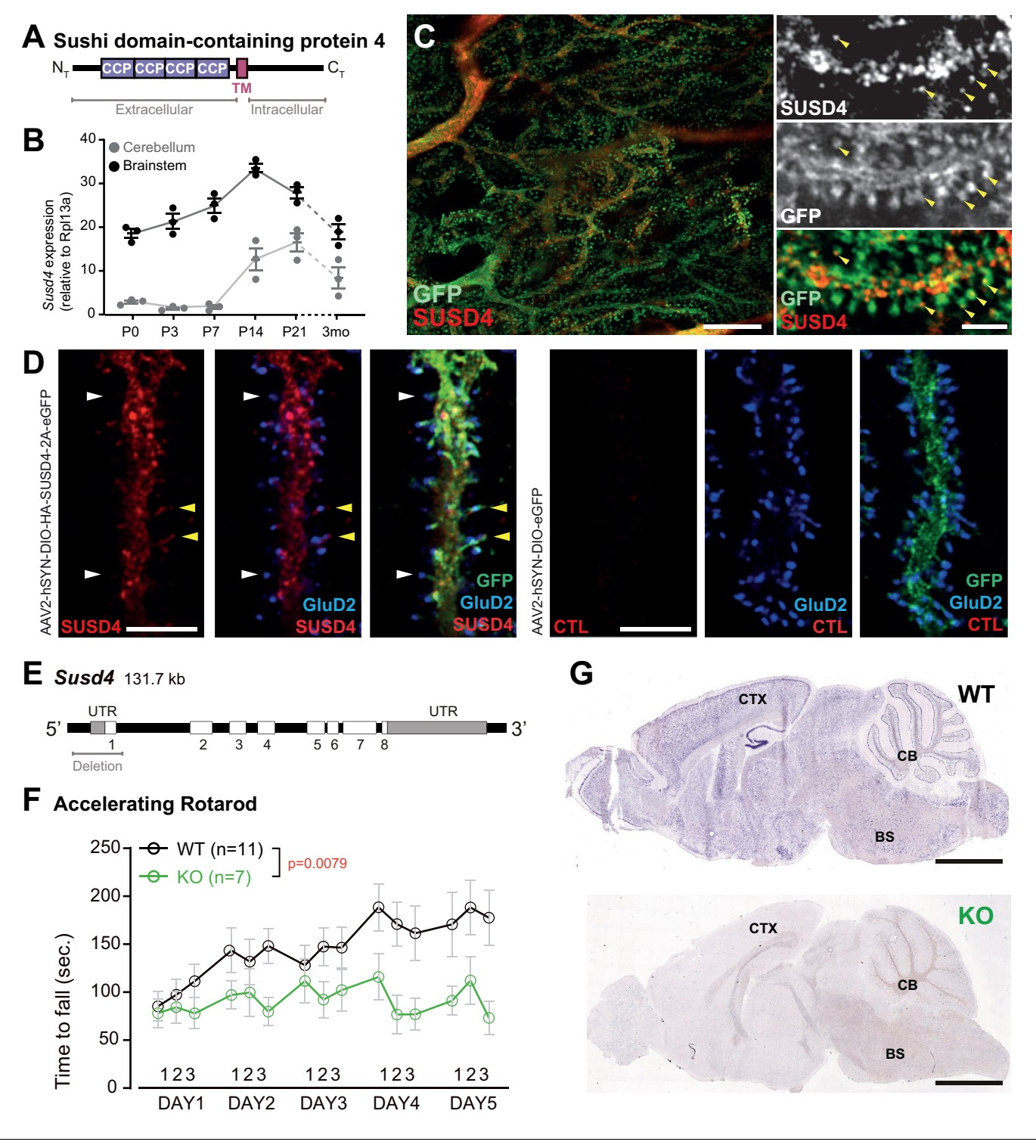

**Figure 1.** SUSD4 is necessary for motor coordination adaptation and learning. (A) Diagram of the protein SUSD4 showing its domain organization with four extracellular Complement Control Protein (CCP) domains, one transmembrane (TM) domain and a cytoplasmic domain ($C_T$). (B) Quantitative RT-PCR shows an increase in *Susd4* mRNA expression (relative to the housekeeping gene *Rpl13a*) during postnatal development in the cerebellum and in the brainstem. Extracts were prepared from tissue samples of mice aged from 0 to 21 days (P0-21) and 3 months (3mo). Mean ± s.e.m. (n = 3 independent experiments). (C) HA-tagged SUSD4 is found in dendrites (left panel, single plane) and in some of the distal dendritic spines (right panel, arrowheads, projection of a 1.95 μm z-stack) in adult cerebellar Purkinje cells. Anti-HA and anti-GFP immunolabeling was performed on parasagittal cerebellar sections obtained from adult L7Cre mice after stereotaxic injection of AAV particles driving the expression of HA-SUSD4 and soluble GFP.

*Figure 1 continued on next page*

*Figure 1 continued*

Scale bars: 10 µm (left panel) and 2 µm (right panel). (D) Purkinje cells from primary mixed cerebellar cultures of L7Cre mice were transduced at 3 days in vitro (DIV3) with an HA-tagged SUSD4 expressing virus (AAV2-hSYN-DIO-HA-SUSD4-2A-eGFP) or with a control virus expressing GFP alone (AAV2-hSYN-DIO-eGFP), and immunostained in non-permeabilizing conditions at DIV17 for HA to localize surface SUSD4 (anti-HA, red), and in permeabilizing conditions to detect the green fluorescent protein (anti-GFP, green) and the endogenous GluD2 subunit (anti-GRID2, blue). Scale bar 5 µm. (E) Genomic structure of the *Susd4* gene. White boxes represent exons. Exon 1 is deleted in the *Susd4* loss-of-function mouse model. See also *Figure 1—figure supplement 2*. (F) Motor coordination and learning is deficient in adult male *Susd4* $^{-/-}$ (knockout [KO]) mice compared to age-matched *Susd4* $^{+/+}$ (wild-type [WT]) littermates. Each mouse was tested three times per day during 5 consecutive days on an accelerating rotarod (4-40 r.p.m. in 10 min) and the time spent on the rotarod was measured. Mean ± s.e.m. (WT n = 11 and KO n = 7 mice, two-way ANOVA with repeated measures, interaction (time and genotype): **p=0.0079, $F_{(14, 224)}$=2.22; time: ****p<0.0001, $F_{(14, 224)}$=3.469; genotype: p=0.0553, $F_{(1, 16)}$=4.272). (G) In situ hybridization experiments were performed on brain sections from 1-month-old *Susd4* WT and *Susd4* KO mice to detect *Susd4* mRNA using a probe encompassing exons 2–5 (see also *Figure 1—figure supplement 2*). *Susd4* expression was found in many regions of the brain in *Susd4* WT mice (see also *Figure 1—figure supplement 1*) including the cerebral cortex (CTX), the cerebellum (CB), and the brainstem (BS). No labeling was found in the brain of *Susd4*KO mice. Scale bar 500 µm.

The online version of this article includes the following source data and figure supplement(s) for figure 1:

**Source data 1.** Numerical data to support graphs in *Figure 1*.

**Figure supplement 1.** *Susd4* mRNA expression in the developing mouse brain.

**Figure supplement 2.** Characterization of *Susd4* knockout (KO) mice.

**Figure supplement 3.** Footprint analysis in *Susd4* knockout (KO) mice.

**Figure supplement 4.** Normal cerebellar cytoarchitecture in *Susd4* knockout (KO) mice.

**Figure supplement 5.** High-density microelectrode array (MEA) analysis of Purkinje cell spiking in acute cerebellar slices from *Susd4* knockout (KO) compared to *Susd4* wild type (WT).

binding complexes identified proteins that are involved in the regulation of several parameters controlling AMPA receptor turnover. We showed that SUSD4 directly interacts with E3 ubiquitin ligases of the NEDD4 family, which are known to regulate ubiquitination and degradation of their substrates. Our results also show that SUSD4 and GluA2 can interact in transfected HEK293 cells and partially colocalize in cultured PCs. Altogether, these findings suggest a function of SUSD4 in the regulation of GluA2 trafficking and degradation allowing proper synaptic plasticity and learning.

## Results

### *Susd4* is broadly expressed in neurons during postnatal development

Given the potential synaptic role for SUSD4, its pattern of expression should correlate with the timing of synapse formation and/or maturation during postnatal development. In situ hybridization experiments using mouse brain sections showed high expression of *Susd4* mRNA in neurons in many regions of the central nervous system, including the cerebral cortex, the hippocampus, the cerebellum, and the brainstem (*Figure 1B* and *Figure 1—figure supplement 1*). *Susd4* expression was already detected as early as postnatal day 0 (P0) in some regions but increased with brain maturation (*Figure 1—figure supplement 1*). In the cerebellum, a structure where the developmental sequence leading to circuit formation and maturation is well described (*Sotelo, 2004*), quantitative RT-PCR showed that *Susd4* mRNA levels start increasing at P7 and by P21 reach about 15 times the levels detected at birth (*Figure 1B*). At P7, a major increase in synaptogenesis is observed in the cerebellum. At this stage, hundreds of thousands of PF excitatory synapses form on the distal dendritic spines of each PC, and a single CF arising from an inferior olivary neuron translocates and forms about 300 excitatory synapses on proximal PC dendrites (*Leto et al., 2016*). In the brainstem, where cell bodies of inferior olivary neurons are located, the increase in *Susd4* mRNA expression occurs earlier, already by P3, and reaches a peak by P14 (*Figure 1B*). Similarly to the cerebellum, this pattern of *Susd4* expression parallels the rate of synaptogenesis that increases during the first postnatal week in the inferior olive (*Gotow and Sotelo, 1987*). To identify the subcellular localization of the SUSD4 protein and because of the lack of suitable antibodies for immunolabeling, viral particles enabling CRE-dependent coexpression of HA-tagged SUSD4 and GFP in neurons were injected in the cerebellum of adult mice expressing the CRE recombinase specifically in cerebellar PCs. Immunofluorescent labeling against the HA tag demonstrated the localization of HA-SUSD4 in dendrites and in some of the numerous dendritic spines present on the surface of distal dendrites (*Figure 1C*).

These spines are the postsynaptic compartments of PF synapses in PCs. Immunofluorescence analysis of transduced cultured PCs showed that HA-tagged SUSD4 could be immunolabeled in non-permeabilizing conditions and located at the surface of dendrites and spines (*Figure 1D*). Double labeling with the postsynaptic marker GluD2 (GRID2) further showed partial colocalization at the surface of some, but not all, spines. Therefore, the timing of *Susd4* mRNA expression during postnatal development and the subcellular localization of the SUSD4 protein in cerebellar PCs are in agreement with a potential role for SUSD4 in excitatory synapse formation and/or function.

## *Susd4* loss-of-function leads to deficits in motor coordination and learning

To determine the synaptic function of SUSD4, we analyzed the phenotype of $Susd4^{-/-}$ constitutive KO mice with a deletion of exon 1 (*Figure 1E and G* and *Figure 1—figure supplement 2*). RT-PCR using primers encompassing the last exons and the 3'UTR show the complete absence of *Susd4* mRNA in the brain of these *Susd4* KO mice (*Figure 1—figure supplement 2*). No obvious alterations of mouse development and behavior were detected in those mutants, an observation that was confirmed by assessment of their physical characteristics (weight, piloerection), basic behavioral abilities such as sensorimotor reflexes (whisker responses, eye blinking), and motor responses (open-field locomotion; see *Supplementary file 1*). We further assessed the behavior of *Susd4* KO mice for motor coordination and motor learning (*Kayakabe et al., 2014*; *Lalonde and Strazielle, 2001*; *Rondi-Reig et al., 1997*). Using a footprint test, a slightly larger print separation of the front and hind paws in the *Susd4* KO mice was detected, but no differences in the stride length and stance width were found (*Figure 1—figure supplement 3*). In the accelerated rotarod assay, a classical test of motor adaptation and learning (*Buitrago et al., 2004*), the mice were tested three times per day at 1 hr interval during 5 consecutive days. The *Susd4* KO mice performed as well as the $Susd4^{+/+}$ (wild-type [WT]) littermate controls on the first trial (*Figure 1F*, day 1, trial 1). This indicates that there is no deficit in their balance function, despite the slight change in fine motor coordination found in the footprint test. However, while the control mice improved their performance as early as the third trial on the first day, and further improved with several days of training, no learning could be observed for the *Susd4* KO mice either during the first day, or in the following days (*Figure 1F*). These results show that *Susd4* loss-of-function leads to impaired motor coordination and learning in adult mice.

## *Susd4* loss-of-function prevents LTD at cerebellar PF/PC synapses

Because of the high expression of *Susd4* in cerebellar PCs (*Figure 1G* and *Figure 1—figure supplement 1*), we focused on this neuronal type to identify the morphological and functional consequences of *Susd4* loss-of-function. No deficits in the global cytoarchitecture of the cerebellum and morphology of PCs were found in *Susd4* KO mice (*Figure 1—figure supplement 4*). Using high-density microelectrode array (MEA), we assessed the spontaneous activity of PCs in acute cerebellar slices from *Susd4* KO mice and compared to *Susd4* WT mice (*Figure 1—figure supplement 5*). No differences were detected in either the mean spiking frequency, the coefficient of variation (CV) of interspike intervals (ISIs), or the intrinsic variability of spike trains (CV2, *Holt and Douglas, 1996*), indicating that the firing properties of PCs are not affected by *Susd4* loss-of-function.

Co-immunolabeling of PF presynaptic boutons using an anti-VGLUT1 antibody and of PCs using an anti-calbindin antibody in cerebellar sections from juvenile WT mice revealed an extremely dense staining in the molecular layer corresponding to the highly numerous PFs contacting PC distal dendritic spines (*Figure 2A*). The labeling pattern appeared to be similar in *Susd4* KO. High-resolution microscopy and quantitative analysis confirmed that there are no significant changes in the mean density and volume of VGLUT1 clusters following *Susd4* loss-of-function (*Figure 2A*). Electric stimulation of increasing intensity in the molecular layer allows the progressive recruitment of PFs (*Konnerth et al., 1990*) and can be used to assess the number of synapses and basic PF/PC transmission using whole-cell patch-clamp recordings of PCs on acute cerebellar slices (*Figure 2B*). No difference was observed in the amplitude and the kinetics of the responses to PF stimulation in PCs from *Susd4* KO and control littermate mice (*Figure 2C* and *Figure 2—figure supplement 1*). Furthermore, the probability of vesicular release in the presynaptic PF boutons, as assessed by measurements of paired-pulse facilitation (*Atluri and Regehr, 1996*; *Konnerth et al., 1990*; *Valera et al.,*

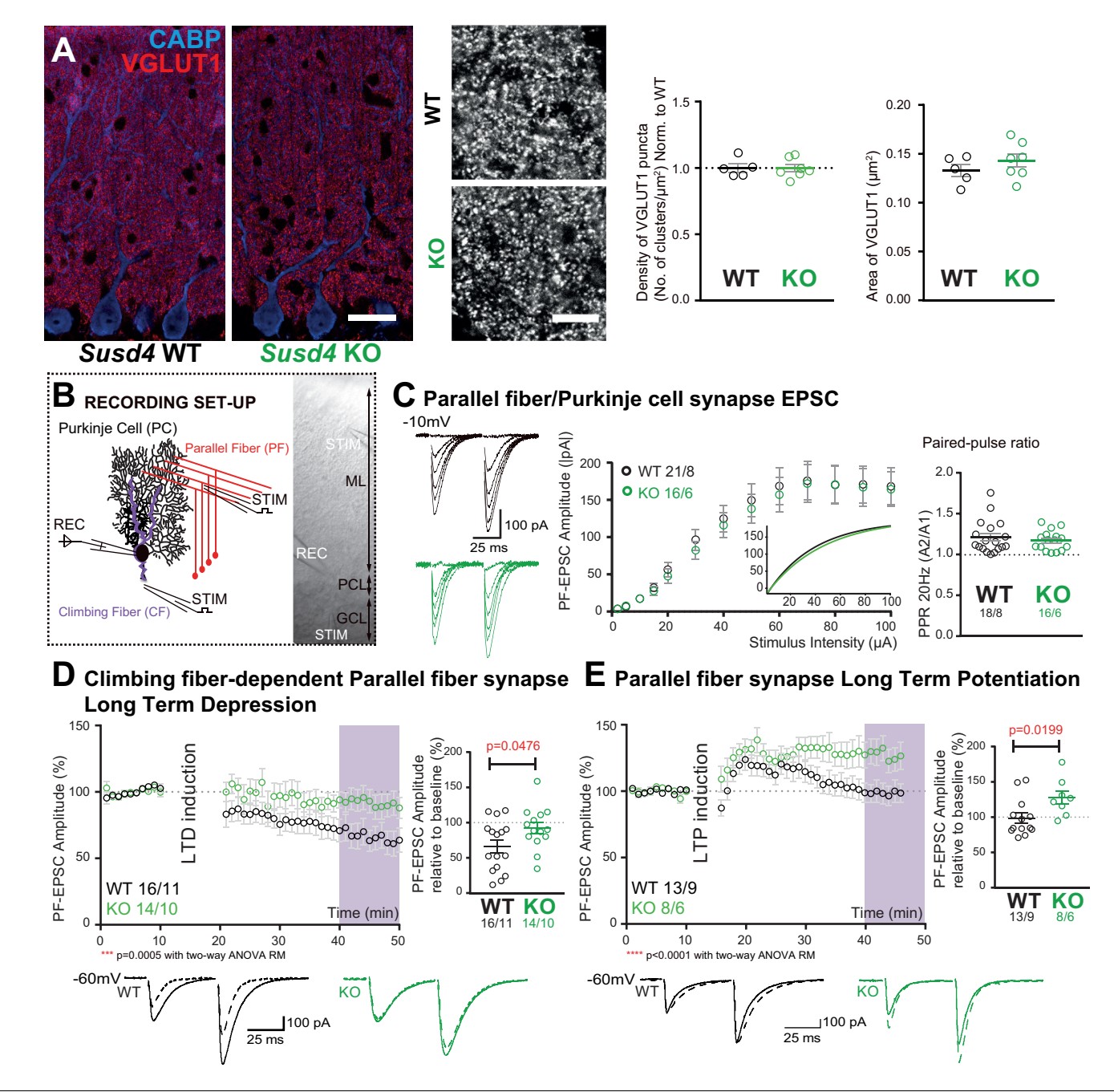

**Figure 2.** *Susd4* loss-of-function leads to deficient long-term depression and facilitated long-term potentiation of parallel fiber (PF)/Purkinje cell synapses. (**A**) Quantitative analysis of the morphology of PF presynaptic boutons immunolabeled by an anti-VGLUT1 antibody (red) in Purkinje cells (anti-CABP, blue). Quantifications of the density and the area of the VGLUT1 clusters did not reveal any difference between *Susd4* knockout (KO) and *Susd4* wild-type (WT) mice. Mean ± s.e.m. (WT n = 5 and KO n = 7 mice; VGLUT1 clusters density: Mann-Whitney test, p>0.9999; area VGLUT1 clusters: unpaired Student's t-test, p=0.3089). Scale bars 30 μm (left) and 10 μm (right). (**B**) Diagram of the setup for patch-clamp recordings (REC) of Purkinje cells in 300-μm-thick parasagittal cerebellar slices. PF and climbing fiber responses were elicited by electrical stimulation (STIM). ML: molecular layer; PCL: Purkinje cell layer; GCL: granule cell layer. (**C**) Input-output curve of the PF/Purkinje cell transmission. The amplitude of the elicited excitatory postsynaptic currents (EPSCs) increases with the intensity of the stimulus and is not significantly different between *Susd4* KO and WT littermates. The fitted curves for each genotype are presented in the inset. Representative sample traces are presented. Mean ± s.e.m. (WT n = 18 cells from eight mice and KO n = 16 cells from six mice; Kolmogorov-Smirnov test, p=0.8793). Short-term plasticity of PF/Purkinje cell synapses is not affected by *Susd4* loss-of-function. PFs were stimulated twice at 50 ms interval and the paired-pulse ratio (PPR) was calculated by dividing the amplitude of the second peak by the amplitude of the first peak. Mean ± s.e.m. (WT n = 21 cells from eight mice and KO n = 16 cells from six mice;

*Figure 2 continued on next page*

*Figure 2 continued*

Mann-Whitney test, p=0.9052). (**D**) Climbing fiber-dependent PF/Purkinje cell synapse long-term depression (LTD) is impaired in the absence of *Susd4* expression. LTD was induced by pairing stimulations of PFs and climbing fibers at 100 ms interval during 10 min at 0.5 Hz (see also ***Figure 2—figure supplement 1***). The amplitude of the PF EPSC was measured using two consecutive PF stimulations at 50 ms interval. Representative sample traces are presented. Right: EPSC amplitudes from the last 10 min (purple) of recordings were used to calculate the LTD ratio relative to baseline. Mean ± s.e.m. (WT n = 16 cells from eleven mice and KO n = 14 cells from ten mice; two-tailed Wilcoxon signed rank test with null hypothesis of 100: WT **p=0.0063; KO p=0.2676; Mann-Whitney test, WT vs. KO *p=0.0476). (**E**) Loss-of-function of *Susd4* facilitates PF/Purkinje cell synapse long-term potentiation (LTP). Tetanic stimulation of only PFs at 0.3 Hz for 100 times (see also ***Figure 2—figure supplement 1***) induced LTP in *Susd4* KO Purkinje cells while inducing only a transient increase in PF transmission in WT Purkinje cells. Representative sample traces are presented. Right: EPSC amplitudes from the last 7 min (purple) were used to calculate the LTP ratio relative to baseline. Mean ± s.e.m. (WT n = 13 cells from nine mice and KO n = 8 cells from six mice; two-tailed Wilcoxon signed rank test with null hypothesis of 100: WT p=0.5879; KO *p=0.0234; Mann-Whitney test, WT vs. KO: *p=0.0199).

The online version of this article includes the following source data and figure supplement(s) for figure 2:

**Source data 1.** Numerical data to support graphs in ***Figure 2***.

**Figure supplement 1.** Parallel fiber (PF)/Purkinje cell (PC) synapse excitatory postsynaptic currents (EPSCs) kinetics, long-term plasticity induction protocols, paired-pulse facilitation ratio, and delayed EPSC quanta.

---

*2012*), was not changed at PF/PC synapses (***Figure 2C***). Finally, no differences in the frequency and amplitude of PF/PC-evoked quantal events were detected (***Figure 2—figure supplement 1***). Thus, in accordance with the morphological analysis, *Susd4* invalidation has no major effect on the number and basal transmission of PF/PC synapses in the mouse.

Long-term synaptic plasticity of PF/PC synapses is involved in proper motor coordination and adaptation learning (***Gutierrez-Castellanos et al., 2017***; ***Hirano, 2018***; ***Kakegawa et al., 2018***). We first assessed LTD in PF/PC synapses using conjunctive stimulation of PFs and CFs and whole-cell patch-clamp recordings of PCs in acute cerebellar slices from juvenile mice. The LTD induction protocol produced a 42% average decrease in the amplitude of PF excitatory postsynaptic currents (EPSCs) in PCs from WT mice while the paired-pulse facilitation ratio was not changed during the course of our recordings (***Figure 2D*** and ***Figure 2—figure supplement 1***). In *Susd4* KO PCs, the same LTD induction protocol did not induce any significant change in PF EPSCs during the 30 min recording period, showing that LTD induction and maintenance are greatly impaired in the absence of SUSD4 (***Figure 2D***). We then assessed LTP induction using high-frequency stimulation of PF in the absence of inhibition blockade as in ***Binda et al., 2016***. In slices from *Susd4* WT mice, tetanic stimulation every 3 s during 5 min induced only a transient increase in transmission of about 20% and the amplitude of the response returned to baseline after only 15 min (***Figure 2E*** and ***Figure 2—figure supplement 1***). This result suggests that under our experimental conditions and in this particular genetic background, LTD might be favored in contrast to previously obtained results (***Binda et al., 2016***; ***Titley et al., 2019***). In the case of *Susd4* KO PCs, the same protocol induced LTP with a 27% increase in transmission that was maintained after 35 min (***Figure 2E***). These results indicate that the absence of *Susd4* expression promoted LTP induction at PF/PC synapses.

Lack of LTD of PF/PC synapses could arise from deficient CF/PC transmission. To test this possibility, we first crossed the *Susd4* KO mice with the *Htr5b*-GFP BAC transgenic line (http://gensat.org/MMRRC_report.jsp?founder_id=17735) expressing soluble GFP specifically in inferior olivary neurons in the olivocerebellar system to visualize CFs. We found that CFs had a normal morphology and translocated along the proximal dendrites of their PC target in *Susd4* KO mice (***Figure 3—figure supplement 1***). We then assessed whether developmental elimination of supernumerary CFs was affected by *Susd4* invalidation using whole-cell patch-clamp recordings of PCs on cerebellar acute slices (***Crepel et al., 1976***; ***Hashimoto and Kano, 2003***). No difference was found in the percentage of remaining multiply-innervated PCs in the absence of *Susd4* (***Figure 3—figure supplement 1***). We next used VGLUT2 immunostaining to label CF presynaptic boutons and analyze their morphology using high-resolution confocal microscopy and quantitative image analysis. VGLUT2 immunostaining revealed the typical CF innervation territory on PC proximal dendrites, extending up to about 80% of the molecular layer height both in control *Susd4* WT and in *Susd4* KO mice (***Figure 3A***). Furthermore, the number and density of VGLUT2 clusters were not significantly different between *Susd4* WT and *Susd4* KO mice. To test whether the lack of CF-dependent PF LTD was due to deficient CF transmission, we used whole-cell patch-clamp recordings of PCs in acute cerebellar slices. Contrary to what could have been expected, the typical all-or-none CF-evoked EPSC

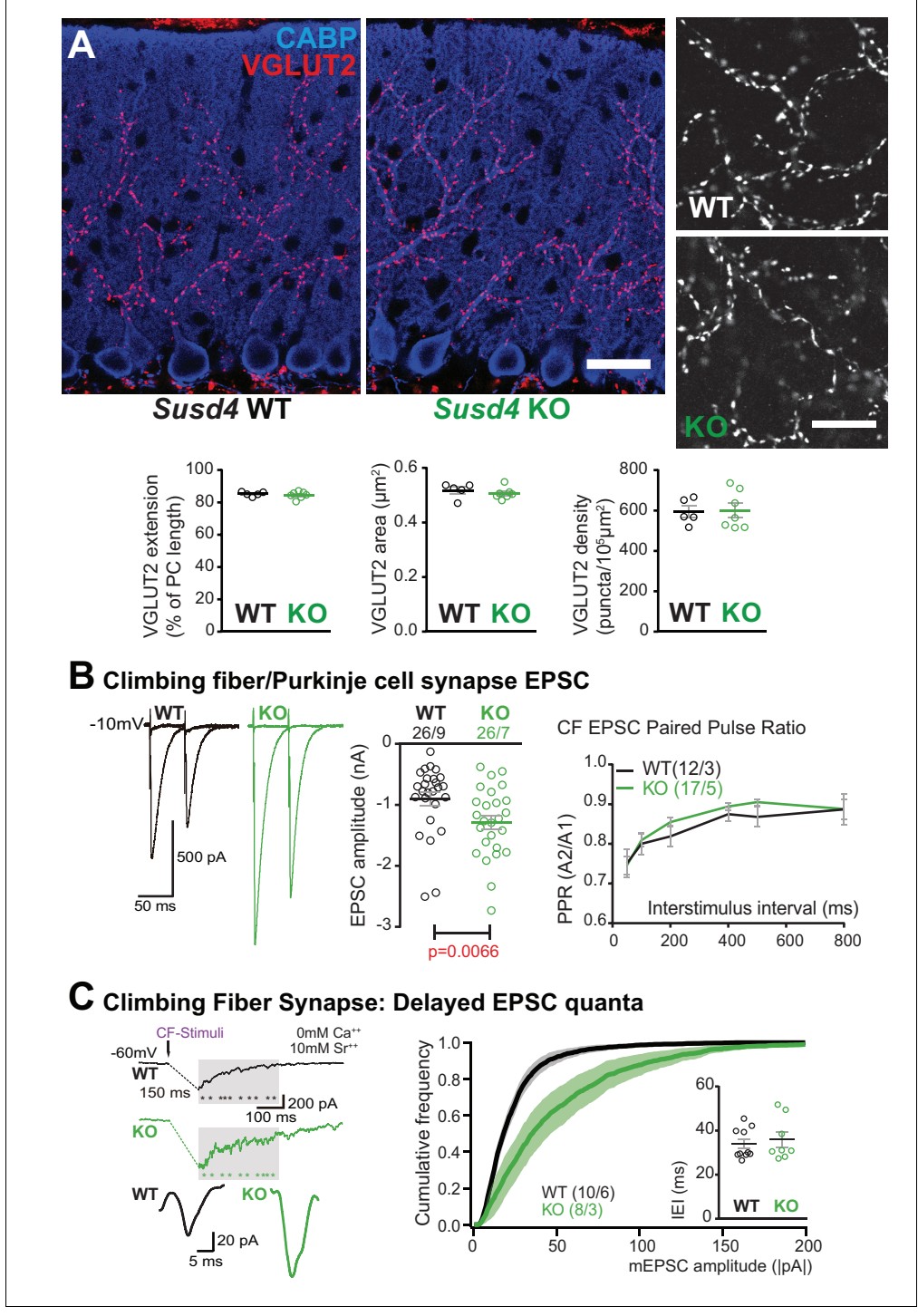

**Figure 3.** Transmission at the climbing fiber (CF)/Purkinje cell synapses is increased in *Susd4* knockout (KO) mice. (**A**) Climbing fiber presynaptic boutons were immunostained with an anti-VGLUT2 antibody in cerebellar sections from P30 *Susd4* wild-type (WT) and *Susd4* KO mice. The extension of the CF synaptic territory was calculated by measuring the extent of the VGLUT2 (red) labeling relative to the height of the Purkinje cell dendritic tree (immunostained using an anti-CABP antibody, blue). Quantification of the mean density of VGLUT2 puncta and their mean area showed no differences between *Susd4* KO mice and their control littermates. Mean ± s.e.m. (WT n = 5 and KO n = 7 mice; VGLUT2 extension: Mann-Whitney test, p=0.6389; VGLUT2 area: unpaired Student's t-test, p=0.4311; VGLUT2 density: unpaired Student's t-test, p=0.8925). Scale bars 30 μm (left) and 10 μm (right). (**B**) Short-term synaptic plasticity of CF/Purkinje cell synapses was elicited by two consecutive stimulations at

*Figure 3 continued on next page*

*Figure 3 continued*

various intervals. The amplitude of the CF-elicited excitatory postsynaptic current (EPSC) was increased in *Susd4* KO mice compared to WT littermates. (WT n = 26 cells, nine mice and KO n = 26 cells, seven mice, Mann-Whitney test, **p=0.0066). No difference in the paired-pulse ratios (PPRs) was detected at any interval between *Susd4* KO mice and WT mice. Representative sample traces are presented. See also *Figure 3—figure supplement 1*. Mean ± s.e.m. (WT n = 12 cells from three mice and KO n = 17 cells from five mice; Kolmogorov-Smirnov test, p=0.4740). (C) Delayed CF-EPSC quanta were evoked by CF stimulation in the presence of $Sr^{++}$ instead of $Ca^{++}$ to induce desynchronization of fusion events. Representative sample traces are presented. The cumulative probability for the amplitude of the events together with the individual amplitude values for each event show an increased amplitude associated with *Susd4* loss-of-function. The individual frequency values for each cell (measured as interevent interval, IEI) present no differences between the genotypes. See also *Figure 3—figure supplement 1*. Mean ± s. e.m. (WT n = 10 cells from six mice and KO n = 8 cells from three mice; amplitude: Kolmogorov-Smirnov distribution test, ***p<0.0001; frequency: Mann-Whitney test, p=0.6334).

The online version of this article includes the following source data and figure supplement(s) for figure 3:

**Source data 1.** Numerical data to support graphs in *Figure 3*.

**Figure supplement 1.** Characteristics of the climbing fiber (CF)/Purkinje cell (PC) synapse.

---

was detected in PCs from *Susd4* KO mice with increased amplitude when compared to WT PCs (*Figure 3B*) while no differences in CF-EPSC kinetics were found (*Figure 3—figure supplement 1*). Analysis of the complex spikes in current-clamp mode during LTD induction did not reveal any change in the complex spike waveform, with the same mean number of spikelets in response to the repeated CF stimulation in *Susd4* WT and *Susd4* KO mice (*Figure 3—figure supplement 1*). Therefore, the lack of CF-dependent PF/PC synapse LTD in *Susd4* KO mice is not due to impaired CF/PC synapse formation or transmission. Measurements of evoked quantal events revealed an increase in the amplitude of the quantal EPSCs at CF/PC synapses from juvenile mice (*Figure 3C* and *Figure 3—figure supplement 1*). Paired-pulse facilitation and depression at PF/PC and CF/PC synapses, respectively, are similar between *Susd4* KO and control mice, both in basal conditions and during plasticity recordings (*Figure 2C*, *Figure 3B*, *Figure 2—figure supplement 1*), suggesting strongly that the changes in PF/PC synaptic plasticity and in CF/PC transmission in *Susd4* KO PCs have a postsynaptic origin. Overall our results show that *Susd4* loss-of-function in mice leads to a highly specific phenotype characterized by misregulation of postsynaptic plasticity in the absence of defects in synaptogenesis and in basal transmission in cerebellar PCs.

## *Susd4* loss-of-function leads to deficient activity-dependent degradation of GluA2

What are the mechanisms that allow regulation of long-term synaptic plasticity by SUSD4? The lack of LTD at PF/PC synapses and our analysis of evoked quantal events suggested the involvement of SUSD4 in the regulation of postsynaptic receptor numbers. GluA2 subunits are present in most AMPA receptor channels in PC excitatory synapses (*Masugi-Tokita et al., 2007*; *Zhao et al., 1998*). To assess whether *Susd4* loss-of-function leads to misregulation of the GluA2 subunits at PC excitatory synapses, we first performed co-immunolabeling experiments using an anti-GluA2 antibody and an anti-VGLUT2 antibody on cerebellar sections followed by high-resolution microscopy. Several GluA2 clusters of varying sizes were detected in close association with each VGLUT2 presynaptic cluster corresponding to a single CF release site, while very small and dense GluA2 clusters were found in the rest of the molecular layer which mostly correspond to GluA2 clusters at the PF/PC synapses (*Figure 4A*). No obvious change in GluA2 distribution in the molecular layer in *Susd4* KO mice was found when compared to controls, in accordance with normal basal transmission in PF/PC synapses (*Figure 2C*). Quantitative analysis of the GluA2 clusters associated with VGLUT2-labeled CF presynaptic boutons did not reveal a significant change in the total mean intensity of GluA2 clusters per CF presynaptic bouton (*Figure 4A*). However, the proportion of CF presynaptic boutons with no GluA2 cluster was smaller in juvenile *Susd4* KO mice than in WT mice (*Figure 4A*). This decrease partially explains the increase in the amplitude of quantal EPSCs and CF transmission (*Figure 3C*).

In cerebellar PCs, regulation of the GluA2 subunits at synapses and of their trafficking is essential for PF LTD (*Chung et al., 2003*; *Xia et al., 2000*). To test whether activity-dependent surface localization of GluA2-containing AMPA receptors is affected by loss of *Susd4*, we set up a biochemical

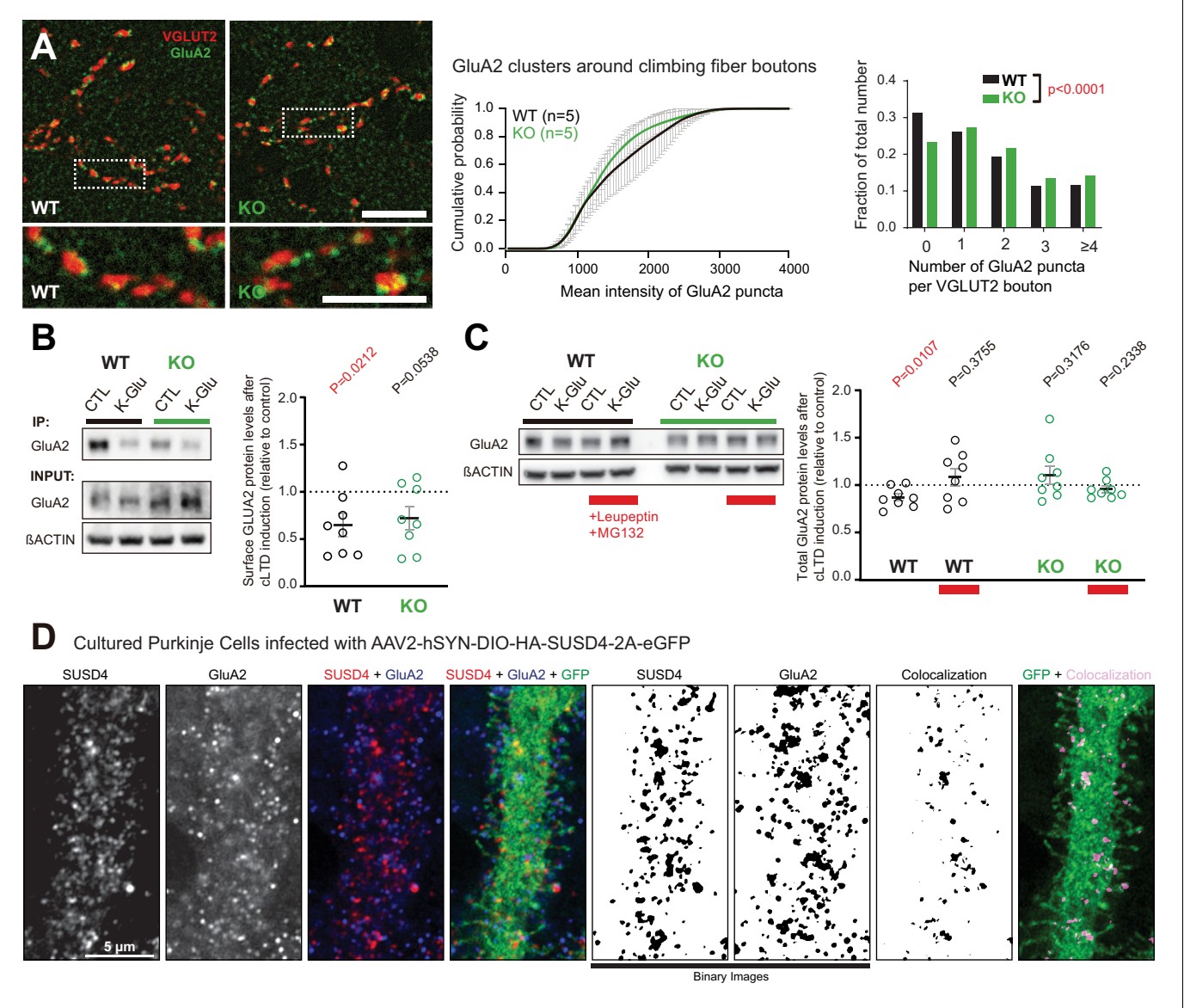

**Figure 4.** Loss of SUSD4 leads to misregulation of the AMPA receptor subunit GluA2. (**A**) The number of GluA2 clusters (anti-GluA2 immunolabeling, green) per climbing fiber presynaptic bouton (anti-VGLUT2 immunolabeling, red) and their intensity were quantified in cerebellar sections of juvenile *Susd4* knockout (KO) mice and *Susd4* wild-type (WT) littermates. Cumulative plot for the mean GluA2 intensity per VGLUT2 bouton shows no significant change between WT and KO. The distribution of the VGLUT2 boutons according to the number of associated GluA2 clusters is significantly different between WT and KO. Mean ± s.e.m. (WT n = 5 and KO n = 5 mice; intensity: Kolmogorov-Smirnov test, p=0.5009; distribution: chi-square contingency test, ****p<0.0001). Scale bars 30 μm (top) and 15 μm (bottom). (**B**) Activity-dependent changes in surface localization of GluA2 was studied in cerebellar acute slices from *Susd4* KO mice and control *Susd4* WT littermates using a chemical LTD protocol (cLTD; K-Glu: K⁺50 mM and glutamate 10 μM for 5 min followed by 30 min of recovery). Surface biotinylation of GluA2 subunits was followed by affinity purification of biotinylated GluA2 subunits and anti-GluA2 immunoblot analysis. The fraction of biotinylated GluA2 was obtained by measuring the levels of biotinylated GluA2 in affinity-purified samples and total GluA2 normalized to βACTIN in input samples for each condition. The ratios between the fraction of biotinylated GluA2 after cLTD and control conditions are represented. Mean ± s.e.m. (n = 8 independent experiments; two-tailed Student's one sample t-test was performed on the ratios with a null hypothesis of 1, $P_{WT}$ = 0.0212 and $P_{KO}$ = 0.0538). (**C**) Activity-dependent degradation of GluA2 was assessed in cerebellar acute slices from *Susd4* KO and WT mice after induction of chemical LTD (cLTD; K-Glu: K⁺50 mM and glutamate 10 μM for 5 min followed by 30 min of recovery). This degradation was absent when slices were incubated with 100 μg/mL leupeptin and with 50 μM MG132 (to inhibit lysosomal and proteasome degradation, respectively), or when slices were obtained from *Susd4* KO mice. Band intensities of GluA2 were normalized to βACTIN. The ratios between levels with cLTD induction (K-Glu) and without cLTD induction (CTL) are represented. See also *Figure 4—figure supplement 1*. Mean ± s.e.m. (n = 8 independent experiments; two-tailed Student's one sample t-test was performed on the ratios with a null hypothesis of 1, $P_{WT}$ = 0.0107, $P_{WT+Leu/MG132}$ = 0.3755, $P_{KO}$ = 0.3176 and $P_{KO+Leu/MG132}$ = 0.2338). (**D**) Purkinje cells from primary cerebellar cultures of L7Cre mice were transduced at 3 days in

*Figure 4 continued on next page*

*Figure 4 continued*

vitro (DIV3) with AAV particles driving the expression of HA-SUSD4 and soluble GFP (AAV2-hSYN-DIO-HA-SUSD4-2A-eGFP) and immunolabeled at DIV17 in non-permeabilizing conditions to localize surface SUSD4 (anti-HA, red) and surface GluA2 subunits (anti-GluA2, blue). Direct green fluorescent protein is shown (GFP, green). Right panels are binarized images of the anti-HA and anti-GluA2 immunolabelings and of the colocalization of these signals (maximum projection of a 1.8 μm z-stack). Scale bar 5 μm.

The online version of this article includes the following source data and figure supplement(s) for figure 4:

**Source data 1.** Numerical data to support graphs in *Figure 4*.
**Figure supplement 1.** Basal surface GluA2 levels and total GluA2 and GluD2 levels in SUSD4 knockout (KO) mice.
**Figure supplement 2.** Interaction and colocalization of HA-SUSD4 and the AMPA receptor subunit GluA2.

assay in which we induced chemical LTD (cLTD) in acute cerebellar slices (*Kim et al., 2017*) and performed surface biotinylation of GluA2 subunits followed by immunoblot quantification. In control conditions, the mean baseline levels of surface GluA2 were not significantly different between *Susd4* WT and *Susd4* KO mice (*Figure 4—figure supplement 1*). As expected, after cLTD a 35% mean reduction of surface GluA2 receptors was measured in slices from WT mice (*Figure 4B*; p=0.0212, two-tailed Student's t test with a null hypothesis of 1). In acute slices from *Susd4* KO mice, a similar, but not statistically significant, mean reduction of surface GluA2 receptors was detected after cLTD (28%; p=0.0538, two-tailed Student's t test with a null hypothesis of 1). Thus, SUSD4 loss-of-function does not lead on average to a major change in the activity-dependent regulation of the number of surface GluA2 subunits.

Another parameter that needs to be controlled for proper LTD in PCs is the total number of AMPA receptors in the recycling pool and the targeting of AMPA receptors to late endosomes and lysosomes (*Kim et al., 2017*). Lack of LTD and facilitation of LTP in *Susd4* KO mice (*Figure 2D and E*) suggest that GluA2 activity-dependent targeting to the endolysosomal compartment and its degradation is affected by *Susd4* loss-of-function. Using our cLTD assay in cerebellar slices, we measured the total GluA2 levels either in control conditions or in the presence of inhibitors of the proteasome (MG132) and of lysosomal degradation (leupeptin). The comparison of the GluA2 levels in the presence of both inhibitors and in control conditions allowed us to estimate the GluA2 degraded pool, regardless of the mechanism behind this degradation. On average, total GluA2 levels were not significantly different between *Susd4* WT and *Susd4* KO cerebellar slices in basal conditions (*Figure 4—figure supplement 1*), in accordance with our morphological and electrophysiological analysis of PF/PC synapses (*Figure 2A and C*). In slices from WT mice, chemical induction of LTD induced a significant reduction of 13% in total GluA2 protein levels (*Figure 4C*). This reduction was prevented by incubation with the mixture of degradation inhibitors, MG132 and leupeptin, showing that it corresponds to the pool of GluA2 degraded in an activity-dependent manner (*Figure 4C*). In slices from *Susd4* KO mice, this activity-dependent degradation of GluA2 was completely absent. Additionally, the chemical induction of LTD had no effect on the total protein levels of GluD2, another synaptic receptor highly present at PF/PC postsynaptic densities, either in slices from WT or from *Susd4* KO mice (*Figure 4—figure supplement 1*). Thus, SUSD4 specifically controls the activity-dependent degradation of GluA2-containing AMPA receptors during LTD.

Finally, co-immunoprecipitation experiments were performed using extracts from heterologous HEK293 cells transfected with SEP-tagged GluA2 and HA-tagged SUSD4 or the transmembrane protein PVRL3α as a control. After affinity purification of SEP-GluA2, HA-SUSD4 was detected in affinity-purified extracts while PVRL3α was not, showing the specific interaction of SEP-GluA2 and HA-SUSD4 in transfected HEK293 cells (*Figure 4—figure supplement 2*). In order to assess the potential colocalization of SUSD4 and GluA2 in neurons, we used a Cre-dependent AAV construct to express HA-tagged SUSD4 in cultured PCs (*Figure 4D*) and performed immunolabeling of surface GluA2 subunits. Clusters of HA-tagged SUSD4 partially colocalize with GluA2 clusters at the surface of some dendritic spines (*Figure 4D*). Partial colocalization of GluA2 and SUSD4 in neurons was also confirmed in transfection experiments in hippocampal neurons (*Figure 4—figure supplement 2*). Thus, SUSD4 could regulate activity-dependent degradation of GluA2-containing AMPA receptors through a direct interaction.

## SUSD4 interacts with NEDD4 ubiquitin ligases

To better understand how SUSD4 regulates the number of GluA2-containing AMPA receptors at synapses, we searched for SUSD4 molecular partners by affinity purification of cerebellar synaptosome extracts using GFP-tagged SUSD4 as a bait (*Figure 5A*). Interacting partners were identified by proteomic analysis using liquid chromatography with tandem mass spectrometry (LC-MS/MS; *Savas et al., 2014*). Twenty-eight candidates were identified including proteins with known function in the regulation of AMPA receptor turnover (*Figure 5E*). Several candidates were functionally linked to ubiquitin ligase activity by gene ontology (GO) term analysis (*Figure 5A* and *Table 1*). In particular, five members of the NEDD4 subfamily of HECT E3 ubiquitin ligases were found as potential interacting partners, three of them (*Nedd4l*, *Wwp1*, and *Itch*) exhibiting the highest enrichment factors among the 28 candidates. Ubiquitination is a post-translational modification essential for the regulation of protein turnover and trafficking in cells (*Tai and Schuman, 2008*). A survey of the expression of HECT-ubiquitin ligases shows that different members of the NEDD4 subfamily are broadly expressed in the mouse brain, however with only partially overlapping patterns (*Figure 5—figure supplement 1*, http://mouse.brain-map.org, Allen Brain Atlas). *Nedd4* and *Wwp1* are the most broadly expressed, including in neurons that also express *Susd4*, such as hippocampal neurons, inferior olivary neurons in the brainstem, and cerebellar PCs. Immunoblot analysis of affinity-purified synaptosome extracts confirmed the interaction of SUSD4 with NEDD4, ITCH, and WWP1 (*Figure 5B*). Removal of the intracellular domain of SUSD4 (SUSD4ΔC$_T$ mutant) prevented this interaction demonstrating the specificity of SUSD4 binding to NEDD4 ubiquitin ligases (*Figure 5B*).

The NEDD4 subfamily of HECT ubiquitin ligases is known to ubiquitinate and target for degradation many key signaling molecules, including GluA1- and GluA2-containing AMPA receptors (*Schwarz et al., 2010*; *Widagdo et al., 2017*). Ubiquitin ligases of the NEDD4 family bind variants of PY motifs on target substrates and adaptors (*Chen et al., 2017*). However, GluA1 and GluA2 subunits lack any obvious motif of this type. In contrast, two potential PY binding sites are present in the intracellular domain of SUSD4 (*Figure 5C*). To test whether SUSD4 and GluA2 interaction is affected by SUSD4 binding to NEDD4 ubiquitin ligases, co-immunoprecipitation experiments were performed on extracts from heterologous HEK293 cells transfected with SEP-tagged GluA2 and various HA-tagged SUSD4 constructs (*Figure 5C and D*). In addition to several deletion constructs of SUSD4, we generated single- and double-point mutants of the two PY motifs in its intracellular tail (*Figure 5C*). Lack of the cytoplasmic domain completely abrogated binding of NEDD4 to SUSD4, confirming the results obtained using synaptosome extracts (*Figure 5D*). Deletion of the N-terminus domain of SUSD4 did not affect NEDD4 binding. Furthermore, while the mutation of the PPxY site in the intracellular tail (SUSD4-ΔPY mutant) abrogated binding of NEDD4 only partially, mutation of the LPxY site (SUSD4-ΔLY mutant) or of both sites (SUSD4-ΔPY/LY mutant) completely prevented the binding to NEDD4 ubiquitin ligases (*Figure 5C and D*). These mutations did not change significantly the level of HA-SUSD4 protein in transfected HEK293 cells suggesting that the degradation of SUSD4 itself is not regulated by binding of NEDD4 ubiquitin ligases (*Figure 5—figure supplement 2*). In accordance with our results obtained using SEP-GluA2 as a bait (*Figure 4—figure supplement 2*), GluA2 was detected in extracts obtained by affinity purification of the HA-tagged full-length SUSD4 (HA-SUSD4), while it was absent if HA-SUSD4 was replaced by a control transmembrane protein, PVRL3α (*Figure 5D* and *Figure 5—figure supplement 2*). Deletion of the extracellular domain (HA-SUSD4ΔN$_T$) or the cytoplasmic domain (HA-SUSD4ΔC$_T$) did not reduce significantly the ability to interact with SEP-GluA2 when compared to HA-SUSD4 (*Figure 5D* and *Figure 5—figure supplement 2*). Strong co-immunoprecipitation of GluA2 was detected in anti-HA affinity-purified extracts from cells expressing the HA-tagged extracellular domain of SUSD4 alone (HA-SUSD4-N$_T$ construct), showing that this domain is sufficient for GluA2 interaction (*Figure 5D* and *Figure 5—figure supplement 2*). Finally using the SUSD4-ΔLY mutant or SUSD4-ΔPY/LY mutant as a bait did not significantly modify the levels of co-immunoprecipitated GluA2 compared to HA-SUSD4, showing that binding of NEDD4 ubiquitin ligases does not affect SUSD4's ability to interact with GluA2.

## Discussion

Our study shows that the CCP domain-containing protein SUSD4 starts to be expressed in various neurons of the mammalian central nervous system when synapses are formed and mature. *Susd4* loss-of-function in mice leads to impaired motor coordination adaptation and learning, misregulation

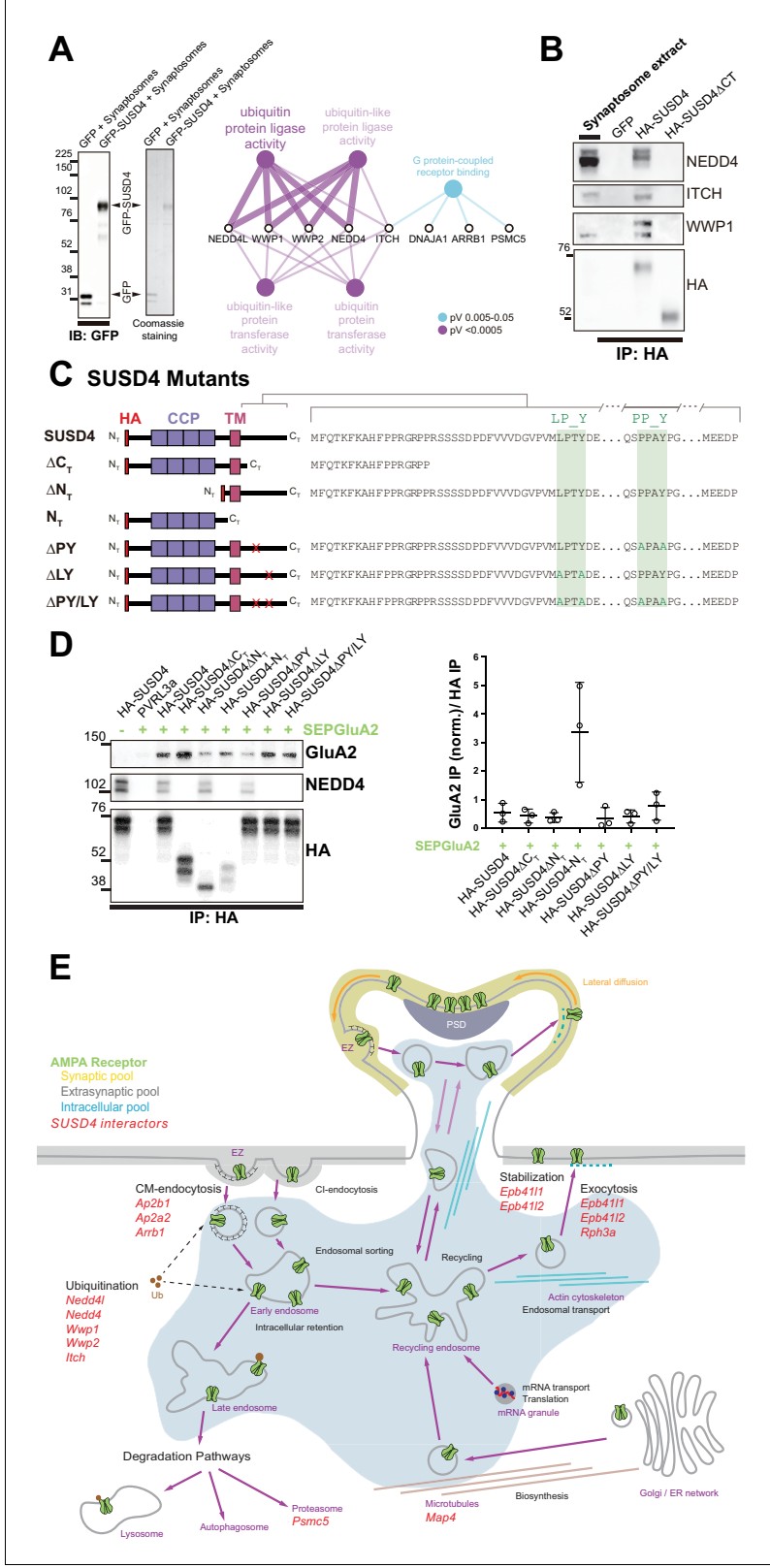

**Figure 5.** SUSD4 binds NEDD4 ubiquitin ligases, known regulators of AMPA receptor turnover and degradation. (**A**) Mass spectrometry identification of SUSD4 interactors. Left: Affinity purification from cerebellar synaptosomes was performed using either GFP-SUSD4 as a bait or GFP as a control. Proteins were then resolved using SDS-PAGE followed by immunoblot for anti-GFP and coomassie staining of proteins. Right: Gene ontology (GO)

*Figure 5 continued on next page*

*Figure 5 continued*

enrichment analysis network (Molecular Function category) of the 28 candidate proteins (Cytoscape plugin ClueGO) identified in affinity purified samples (A) by liquid chromatography with tandem mass spectrometry (LC MS/MS). The ubiquitin ligase activity term is significantly enriched in particular due to the identification of several members of the NEDD4 family of HECT ubiquitin ligases. See also *Table 1* (n = 3 independent experiments). (B) Immunoblot confirmation of SUSD4 interaction with NEDD4 ubiquitin ligases. Affinity purification from cerebellar synaptosomes was performed using full-length SUSD4 (HA-tagged, HA-SUSD4), a mutant lacking the C-terminal tail (HA-SUSD4ΔC$_T$), or GFP as a bait. Proteins were then resolved using SDS-PAGE followed by immunoblot for NEDD4, ITCH, WWP1, or HA-SUSD4 (anti-HA). HA-SUSD4 interacts with all three members of the NEDD4 family. This interaction is lost when the C-terminal tail of SUSD4 is deleted or when GFP is used instead of SUSD4 as a control. (C) Schematic representation of HA-tagged SUSD4 and different mutant constructs: SUSD4ΔC$_T$ (lacking the cytoplasmic tail), SUSD4ΔN$_T$ (lacking the extracellular domain), SUSD4N$_T$ (lacking the transmembrane and intracellular domains), SUSD4ΔPY (point mutation of the PPxY site), SUSD4ΔLY (point mutation of the LPxY), and SUSD4ΔPY/LY (double mutant at both PPxY and LPxY). (D) SUSD4 interaction with GluA2 and NEDD4 was assessed by co-immunoprecipitation using HEK293 cells transfected with SEP-GluA2 together with PVRL3α as a control or one of the HA-SUSD4 constructs represented in (C). Affinity purification was performed with an anti-HA antibody and extracts were probed for co-immunoprecipitation of GluA2 (with an anti-GluA2 antibody) and of the HECT ubiquitin ligase NEDD4 (anti-NEDD4 antibody). Co-immunoprecipitated GluA2 levels are normalized to input GluA2 and then represented as relative to the immunoprecipitated levels for each SUSD4 construct. N = 3 independent experiments. (E) Potential interactors of SUSD4 control several parameters of AMPA receptor turnover. Three different pools of AMPA receptors are found in dendrites and spines: synaptic, extrasynaptic, and intracellular. AMPA receptors are synthetized and delivered close to the synaptic spine to reach the synaptic surface. At the surface, AMPA receptors can move laterally (lateral diffusion) or vertically by endocytosis and exocytosis. Endocytosis can be mediated by clathrin (CM-endocytosis) or be clathrin-independent (CI-endocytosis). CM-endocytosis is often related to activity-dependent processes. After endocytosis, AMPA receptors can choose between two different pathways from the early endosomes, one for recycling and the other for degradation. Potential molecular partners of SUSD4 identified by our proteomic analysis could regulate AMPA receptor turnover at several levels of this cycle (in red).

The online version of this article includes the following source data and figure supplement(s) for figure 5:

**Source data 1.** Numerical data to support graphs in *Figure 5*.
**Figure supplement 1.** Expression of HECT ubiquitin ligases in adult mouse brain.
**Figure supplement 2.** Total protein levels in HEK293 cells transfected with SEP-GluA2 and different SUSD4 mutant constructs (related to *Figure 5C and D*).

of synaptic plasticity in cerebellar PCs, and perturbed degradation of GluA2-containing AMPA receptors after chemically induced LTD. SUSD4 and the GluA2 AMPA receptor subunit interact in transfected heterologous cells and colocalize partially in transduced cultured neurons. Finally, we show that SUSD4 directly binds to ubiquitin ligases of the NEDD4 family, which have been previously shown to regulate GluA2 degradation.

## SUSD4 promotes long-term synaptic depression

The choice between recycling of AMPA receptors to the membrane or targeting to the endolysosomal compartment for degradation is key for the regulation of the number of AMPA receptors at synapses, as well as for the direction and degree of activity-dependent synaptic plasticity (*Ehlers, 2000*; *Lee et al., 2002*). Blocking the trafficking of AMPA receptors through recycling endosomes, for example, using a RAB11 mutant, prevents LTP in neurons (*Park et al., 2004*). Conversely, blocking the sorting of AMPA receptors to the endolysosomal compartment, for example, using a RAB7 mutant, impairs LTD in hippocampal CA1 pyramidal neurons and cerebellar PCs (*Fernández-Monreal et al., 2012*; *Kim et al., 2017*). Further support for the role of receptor degradation comes from mathematical modeling showing that in cerebellar PCs LTD depends on the regulation of the total pool of glutamate receptors (*Kim et al., 2017*). The GluA2 AMPA receptor subunit, and its regulation, is of particular importance for LTD (*Diering and Huganir, 2018*). Phosphorylation in its C-terminal tail and the binding of molecular partners such as PICK1 and GRIP1/2 are known to regulate endocytosis and recycling (*Bassani et al., 2012*; *Chiu et al., 2017*; *Fiuza et al., 2017*), and mutations in some of the phosphorylation sites lead to impaired LTD (*Chung et al., 2003*). The molecular partners regulating the targeting for degradation of GluA2 subunits in an activity-

**Table 1.** List of SUSD4 interactors.

Proteomic identification of SUSD4 interacting partners affinity-purified from synaptosomes extracts using GFP-SUSD4 as a bait (≥2 unique peptides; enrichment factor ≥4).

| UniProtKB accession num. | Protein name | Gene name | Mol. weight (kDa) | Unique peptides | MS/MS count | Enrichment factor |
|---|---|---|---|---|---|---|
| Q8CFI0 | E3 ubiquitin-protein ligase NEDD4-like | Nedd4l | 115,42 | 28 | 319 | 159.5 |
| Q8BH32 | Sushi domain-containing protein 4 | Susd4 | 53,796 | 4 | 97 | 48.5 |
| Q8BZZ3 | NEDD4-like E3 ubiquitin-protein ligase WWP1 | Wwp1 | 104,69 | 13 | 90 | 45 |
| Q8C863 | E3 ubiquitin-protein ligase Itchy | Itch | 98,992 | 24 | 83 | 41.5 |
| Q3TXU5 | Deoxyhypusine synthase | Dhps | 40,642 | 9 | 81 | 40.5 |
| Q9DBG3 | AP-2 complex subunit beta | Ap2b1 | 104,58 | 9 | 47 | 23.5 |
| P50171 | Estradiol 17-beta-dehydrogenase 8 | Hsd17b8 | 26,588 | 2 | 32 | 16 |
| Q9DBH0 | NEDD4-like E3 ubiquitin-protein ligase WWP2 | Wwp2 | 98,76 | 14 | 31 | 15.5 |
| Q922R8 | Protein disulfide-isomerase A6 | Pdia6 | 48,1 | 8 | 26 | 13 |
| P27773 | Protein disulfide-isomerase A3 | Pdia3 | 56,678 | 12 | 24 | 12 |
| P17427 | AP-2 complex subunit alpha-2 | Ap2a2 | 104,02 | 7 | 23 | 11.5 |
| Q8BWG8 | Beta-arrestin-1 | Arrb1 | 46,972 | 4 | 23 | 11.5 |
| Q91WC3 | Long-chain fatty acid – CoA ligase 6 | Acsl6 | 78,016 | 11 | 22 | 11 |
| P27546 | Microtubule-associated protein 4 | Map4 | 117,43 | 9 | 18 | 9 |
| Q505F5 | Leucine-rich repeat-containing protein 47 | Lrrc47 | 63,589 | 9 | 17 | 8.5 |
| Q9Z2H5 | Band 4.1-like protein 1 | Epb41l1 | 98,314 | 8 | 17 | 8.5 |
| P46935 | E3 ubiquitin-protein ligase NEDD4 | Nedd4 | 102,71 | 7 | 17 | 8.5 |
| Q8BMK4 | Cytoskeleton-associated protein 4 | Ckap4 | 63,691 | 11 | 16 | 8 |
| P47708 | Rabphilin-3A | Rph3a | 75,488 | 7 | 15 | 7.5 |
| P42128 | Forkhead box protein K1 | Foxk1 | 74,919 | 6 | 15 | 7.5 |
| P62812 | Gamma-aminobutyric acid receptor subunit alpha-1 | Gabra1 | 51,753 | 7 | 14 | 7 |
| Q60737 | Casein kinase II subunit alpha | Csnk2a1 | 45,133 | 7 | 13 | 6.5 |
| Q99KV1 | DnaJ homolog subfamily B member 11 | Dnajb11 | 40,555 | 5 | 10 | 5 |
| P63037 | DnaJ homolog subfamily A member 1 | Dnaja1 | 44,868 | 4 | 10 | 5 |
| Q9QY76 | Septin-11 | Sept11 | 49,694 | 5 | 9 | 4.5 |
| O70318 | Band 4.1-like protein 2 | Epb41l2 | 109,94 | 6 | 8 | 4 |
| P62196 | 26S protease regulatory subunit 8 | Psmc5 | 45,626 | 5 | 8 | 4 |
| Q9Z2Q6 | Septin-5 | Sept5 | 42,747 | 4 | 8 | 4 |

dependent manner during LTD remain to be identified. Our study shows that loss-of-function of *Susd4* leads both to loss of LTD and loss of activity-dependent degradation of GluA2 subunits. Loss-of-function of *Susd4* does not affect degradation of another postsynaptic receptor, GluD2, showing the specificity of SUSD4 action. Furthermore, loss-of-function of *Susd4* facilitates LTP of PF/PC synapses. Overall our results suggest a role for SUSD4 in the targeting of GluA2-containing AMPA receptors to the degradation compartment during synaptic plasticity.

## SUSD4 interacts with regulators of AMPA receptor turnover

The degradation of specific targets such as neurotransmitter receptors must be regulated in a stimulus-dependent and synapse-specific manner in neurons, to ensure proper long-term synaptic plasticity, learning, and memory (*Tai and Schuman, 2008*). How is this level of specificity achieved?

Adaptor proteins, such as GRASP1, GRIP1, PICK1, and NSF, are known to promote AMPA receptor recycling and LTP (*Anggono and Huganir, 2012*). Such adaptors for the promotion of LTD remain to be found.

Our results show that SUSD4 directly binds to HECT E3 ubiquitin ligases of the NEDD4 family. The family of HECT E3 ubiquitin ligases contains 28 enzymes including the NEDD4 subfamily that is characterized by an N-terminal C2 domain, several WW domains, and the catalytic HECT domain (*Weber et al., 2019*). This subgroup of E3 ligases adds K63 ubiquitin chains to their substrate, a modification that promotes sorting to the endolysosomal compartment for degradation (*Boase and Kumar, 2015*). NEDD4 E3 ligases are highly expressed in neurons in the mammalian brain and have many known substrates with various functions, including ion channels and the GluA1 AMPA receptor subunit. Accordingly, KO mice for the *Nedd4-1* gene die during late gestation (*Kawabe et al., 2010*). The activity and substrate selectivity of NEDD4 E3 ligases thus need to be finely tuned. Both GluA1 and GluA2 AMPA receptor subunits are ubiquitinated on lysine residues in their intracellular tails in an activity-dependent manner (*Lin et al., 2011*; *Lussier et al., 2011*; *Schwarz et al., 2010*; *Widagdo et al., 2015*). Mutation of these lysine residues decreases localization of GluA1 and GluA2 AMPA receptor subunits in the endolysosomal compartment in neurons (*Widagdo et al., 2015*). However, GluA1 and GluA2 subunits lack any obvious intracellular direct binding motif to the WW domain of NEDD4 ubiquitin ligases, raising questions about the precise mechanism allowing regulation of AMPA subunits trafficking and degradation by these enzymes. We showed that SUSD4 and GluA2 AMPA receptor subunits interact in cells and partially colocalize in neurons. SUSD4 could thus regulate the targeting of NEDD4 ubiquitin ligases to AMPA receptors in an activity-dependent manner in neurons. Alternatively, the interaction of SUSD4 with NEDD4 ubiquitin ligases might regulate the trafficking of the SUSD4/GluA2 complex to the degradation pathway. Furthermore, among the potential partners of SUSD4 identified by our proteomics analysis, several other candidates have functions that are relevant for the regulation of synaptic plasticity, such as receptor anchoring, clathrin-mediated endocytosis, and proteasome function (*Figure 5E*). Further work is needed to determine the precise mechanism of action of SUSD4 in neurons in the context of synaptic plasticity.

## SUSD4 and neurodevelopmental disorders

*Susd4* loss-of-function leads to motor impairments, a symptom that is also found in ASD patients (*Fournier et al., 2010*). Deficits in LTD such as the one found in the *Susd4* KO mice are a common feature of several mouse models of ASDs (*Auerbach et al., 2011*; *Baudouin et al., 2012*; *Piochon et al., 2014*). Because of the broad expression of SUSD4 and of ubiquitin ligases of the NEDD4 subfamily in the mammalian central nervous system, whether motor impairments in the *Susd4* KO mice are directly the results of synaptic deficits in cerebellar PCs remain to be demonstrated. Very recently, a reduction in exploratory behavior, in addition to impairments of motor coordination, was reported after *Susd4* loss-of-function (*Zhu et al., 2020*). Thus, mutations in the *Susd4* gene might contribute to the etiology of neurodevelopmental disorders by impairing synaptic plasticity at many synapse types.

In humans, the 1q41-42 deletion syndrome is characterized by many symptoms including IDs and seizures, and in a high majority of the cases, the microdeletion encompasses the *SUSD4* gene (*Rosenfeld et al., 2011*). A *SUSD4* copy number variation has been identified in a patient with ASD (*Cuscó et al., 2009*). *SUSD4* was recently identified among the 124 genes with genome-wide significance for de novo mutations in a cohort of more than 10,000 patients with ASD or IDs (*Coe et al., 2019*). The *GRIA2* gene (coding for the GluA2 subunit) has been found as an ASD susceptibility gene (*Salpietro et al., 2019*; *Satterstrom et al., 2020*), and mutations or misregulation of ubiquitin ligases have been found in many models of ASDs or IDs (*Cheon et al., 2018*; *Lee et al., 2018*; *Satterstrom et al., 2020*). For example, ubiquitination of GluA1 by NEDD4-2 is impaired in neurons from a model of Fragile X syndrome (*Lee et al., 2018*). Understanding the molecular mechanism linking activity-dependent degradation of GluA2 and the SUSD4/NEDD4 complex will thus be of particular importance for our understanding of the etiology of these neurodevelopmental disorders.

# Materials and methods

**Key resources table**

| Reagent type (species) or resource | Designation | Source or reference | Identifiers | Additional information |
|---|---|---|---|---|
| Gene (*Mus musculus*) | *Susd4* | NCBI | Gene ID: 96935 | chr1:182,764,895–182,896,591 |
| Strain (*Mus musculus*) | *Susd4* knockout mice | Lexicon Genetics Incorporated, *Tang et al., 2010* | B6:129S5-Susd4$^{tm1Lex}$ | |
| Strain (*Mus musculus*) | Htr5b-GFP mouse line | Gene Expression Nervous System Atlas (GENSAT) Project | STOCK Tg (Htr5b-EGFP)BZ265Gsat/Mmmh | |
| Strain (*Mus musculus*) | L7Cre mouse line | Jackson Laboratories | B6.129-Tg(Pcp2-cre)2Mpin/J | Stock number 004146 |
| Cell line (*Homo sapiens*) | HEK293H | Gibco | Cat# 11631–017 | |
| Cell line (*Homo sapiens*) | HeLA | Sigma | Cat# 93021013 | |
| Antibody | Mouse monoclonal anti-CABP | Swant | Cat# 300 | (1:1000) |
| Antibody | Rabbit polyclonal anti-CABP | Swant | Cat# CB38 | (1:1000) |
| Antibody | Mouse monoclonal anti-GFP | Abcam | Cat# ab1218 | (1:1000) |
| Antibody | Rabbit polyclonal anti-GFP | Abcam | Cat# ab6556 | (1:1000) |
| Antibody | Mouse monoclonal anti-GLUA2, clone 6C4 | Millipore and BD | Cat# MAB397 and Cat# 556341 | (1:500) |
| Antibody | Rabbit monoclonal anti-GLUA2 | Abcam | Cat# ab206293 | (1:1000) |
| Antibody | Rabbit polyclonal anti-GLURδ1/2 | Millipore | Cat# AB2285 | (1:1000) |
| Antibody | Rat monoclonal anti-HA | Roche Life | Cat# 11867423001 | (1:1000) |
| Antibody | Rabbit monoclonal anti-ITCH | Cell Signaling Technology | Cat# 12117 | (1:1000) |
| Antibody | Rabbit polyclonal anti-NEDD4 | Millipore | Cat# 07–049 | (1:100,000) |
| Antibody | Guinea pig polyclonal anti-VGLUT1 | Millipore | Cat# AB5905 | (1:5000) |
| Antibody | Guinea pig polyclonal anti-VGLUT2 | Millipore | Cat# AB2251 | (1:5000) |
| Antibody | Rabbit polyclonal anti-WWP1 | Proteintech | Cat# 13587–1-AP | (1:2000) |

*Continued on next page*

*Continued*

| Reagent type (species) or resource | Designation | Source or reference | Identifiers | Additional information |
| --- | --- | --- | --- | --- |
| Antibody | Donkey polyclonal anti-goat Alexa Fluor 568 | Invitrogen | Cat# A11057 | (1:1000) |
| Antibody | Donkey anti-mouse Alexa Fluor 488 | Invitrogen | Cat# R37114 | (1:1000) |
| Antibody | Donkey polyclonal anti-mouse Alexa Fluor 568 | Invitrogen | #A10037 | (1:1000) |
| Antibody | Donkey polyclonal anti-rabbit Alexa Fluor 488 | Invitrogen | Cat# A21206 | (1:1000) |
| Antibody | Donkey polyclonal anti-rat Alexa Fluor 594 | Invitrogen | #A21209 | (1:1000) |
| Antibody | Donkey polyclonal anti-rat Alexa Fluor 568 | Abcam | Cat# 175475 | (1:1000) |
| Antibody | Goat polyclonal anti-guinea Pig Alexa Fluor 488 | Invitrogen | Cat# A110-73 | (1:1000) |
| Antibody | Goat polyclonal anti-guinea Pig Alexa Fluor 647 | Invitrogen | Cat# A21450 | (1:1000) |
| Antibody | Goat polyclonal anti-mouse HRP | Jackson Immune Research Laboratories | Cat# 115-035-174 | (1:10,000) |
| Antibody | Goat polyclonal anti-rat HRP | Jackson Immune Research Laboratories | #112-035-175 | (1:10,000) |
| Antibody | Sheep polyclonal anti-digoxigenin alkaline phosphatase | Roche Life Science | Cat# 11093274910 | (1:2000 - 1:5000) |
| Antibody | Mouse monoclonal anti-βACTIN HRP, clone AC-15 | Abcam | Cat# ab49900 | (1:25,000) |
| Recombinant DNA reagent | pHA-SUSD4-GFP | This paper | | From pEGFP-N1 (Addgene, Cat# 6085–1) |
| Recombinant DNA reagent | pHA-SUSD4 | This paper | | |
| Recombinant DNA reagent | pHA-SUSD4-$\Delta$N$_T$ | This paper | | |
| Recombinant DNA reagent | pHA-SUSD4-N$_T$ | This paper | | |
| Recombinant DNA reagent | HA-SUSD4-$\Delta$PY | This paper | | |
| Recombinant DNA reagent | HA-SUSD4-$\Delta$LY | This paper | | |

*Continued on next page*

*Continued*

| Reagent type (species) or resource | Designation | Source or reference | Identifiers | Additional information |
|---|---|---|---|---|
| Recombinant DNA reagent | HA-SUSD4-ΔPY/LY | This paper | | |
| Recombinant DNA reagent | pIRES2-eGFP | Addgene | Cat# 6029–1 | |
| Recombinant DNA reagent | pCAG-PVRL3α | This paper | | From pCAG-mGFP (Addgene, Cat# 14757) |
| Sequenced-based reagent | Susd4_WT_F | This paper | PCR primers | CTG TGG TTT CAA CTG GCG CTG TG |
| Sequenced-based reagent | Susd4_WT_R | This paper | PCR primers | GCT GCC GGT GGG TGT GCG AAC CTA |
| Sequenced-based reagent | Susd4_KO_F | This paper | PCR primers | TTG GCG GTT TCG CTA AAT AC |
| Sequenced-based reagent | Susd4_KO_R | This paper | PCR primers | GGA GCT CGT TAT CGC TAT GAC |
| Sequenced-based reagent | Htr5b-GFP_F | | PCR primers | TTG GCG CGC CTC CAA CAG GAT GTT AAC AAC |
| Sequenced-based reagent | Htr5b-GFP_R | | PCR primers | CGC CCT CGC CGG ACA CGC TGA AC |
| Sequenced-based reagent | L7cre_1 | | PCR primers | GGT GAC GGT CAG TAA ATT GGA C |
| Sequenced-based reagent | L7cre_2 | | PCR primers | CAC TTC TGA CTT GCA CTT TCC TTG G |
| Sequenced-based reagent | L7cre_3 | | PCR primers | TTC TTC AAG CTG CCC AGC AGA GAG C |
| Chemical compound, drug | Picrotoxin | Sigma-Aldrich | Cat# P1675 | |
| Chemical compound, drug | D-AP5 | Tocris | Cat# 0106 | |
| Chemical compound, drug | CGP52432 | Tocris | Cat# 1246 | |
| Chemical compound, drug | JNJ16259685 | Tocris | Cat# 2333 | |
| Chemical compound, drug | DPCPX | Tocris | Cat# 0439 | |
| Chemical compound, drug | AM251 | Tocris | Cat# 1117 | |
| Software, algorithm | Sinaptiqs | Antoine Valera | Software written in Python | http://synaptiqs.wixsite.com/synaptiqs |
| Other | Hoechst 33342 | Sigma | Cat# 14533 | |
| Recombinant viral particles | hSYN-DIO-HA-SUSD4-2A-eGFP-WPRE | Vector biolabs | AAV2 particles | |

## Animals

Susd4 KO mice were generated using 129S5/SvEvBrd ES microinjected in C57BL/6J blastocysts and maintained on the C57BL/6J background (generated by Lexicon Genetics Incorporated, The Woodlands, TX) (*Tang et al., 2010*). Out of the eight *Susd4* exons, coding exon 1 (NCBI accession NM_144796.2) and the 5'UTR (NCBI accession BM944003) were targeted by homologous recombination. This resulted in the deletion of a 1.3 kb sequence spanning the transcription initiation site and exon 1 (*Figure 1E* and *Figure 1—figure supplement 2*). Subsequent genotyping of mice was performed using PCR to detect the WT allele (forward primer: 5' CTG TGG TTT CAA CTG GCG C

TG TG 3'; reverse primer: 5' GCT GCC GGT GGG TGT GCG AAC CTA 3') or the targeted allele (forward primer: 5' TTG GCG GTT TCG CTA AAT AC 3'; reverse primer: 5' GGA GCT CGT TAT CGC TAT GAC 3'). Heterozygous *Susd4*$^{+/-}$ mice were bred to obtain all the genotypes needed for the experiments (*Susd4*$^{+/+}$[WT] and *Susd4*$^{-/-}$[KO] mice) as littermates.

The Htr5b-GFP mouse line was used for labeling of CFs (The Gene Expression Nervous System Atlas [GENSAT] Project, NINDS Contracts N01NS02331 and HHSN271200723701C to The Rockefeller University [New York, NY]). Genotyping was performed using the following primers: 5' TTG GCG CGC CTC CAA CAG GAT GTT AAC AAC 3' and 5' CGC CCT CGC CGG ACA CGC TGA AC 3'.

The L7Cre mouse line was obtained from Jackson laboratories (B6.129-Tg(Pcp2-cre)2Mpin/J; Stock Number: 004146) and genotyping was performed using the following primers: 5' GGT GAC GGT CAG TAA ATT GGA C 3', 5' CAC TTC TGA CTT GCA CTT TCC TTG G 3', and 5' TTC TTC AAG CTG CCC AGC AGA GAG C 3'.

All animal protocols were approved by the *Comité Regional d'Ethique en Experimentation Animale* (no. 00057.01) and animals were housed in authorized facilities of the CIRB (# C75 05 12).

## Antibodies

The following primary antibodies were used: mouse monoclonal anti-CABP (1:1000; Swant, Switzerland, Cat#300), rabbit polyclonal anti-CABP (1:1000; Swant, Cat#CB38), mouse monoclonal anti-GFP (1:1000; Abcam, Cambridge, UK, Cat#ab1218), rabbit polyclonal anti-GFP (1:1000; Abcam, Cat#ab6556), mouse monoclonal anti-GluA2 (clone 6C4; 1:500; Millipore, MA, Cat#MAB397 and BD, NJ, Cat#556341), rabbit monoclonal anti-GluA2 (1:1000; Abcam, Cat#ab206293), rabbit polyclonal anti-GluRδ1/2 (1:1000; Millipore, Cat#AB2285), rat monoclonal anti-HA (1:1000; Roche Life Science, Penzberg, Germany, Cat#11867423001), rabbit monoclonal anti-ITCH (1:1000; Cell Signaling Technology, MA, Cat#12117), rabbit polyclonal anti-NEDD4 (1:10,000; Millipore, Cat#07–049), guinea pig polyclonal anti-VGLUT1 (1:5000; Millipore, Cat#AB5905), guinea pig polyclonal anti-VGLUT2 (1:5000; Millipore, Cat#AB2251), and rabbit polyclonal anti-WWP1 (1:2000; Proteintech, Chicago, IL, Cat#13587–1-AP).

The following secondary antibodies were used: donkey polyclonal anti-goat Alexa Fluor 568 (1:1000; Invitrogen, CA, Cat#A11057), donkey anti-mouse Alexa Fluor 488 (1:1000; Invitrogen, Cat#R37114), donkey polyclonal anti-mouse Alexa Fluor 568 (1:1000; Invitrogen, #A10037), donkey polyclonal anti-rabbit Alexa Fluor 488 (1:1000; Invitrogen, Cat#A21206), donkey polyclonal anti-rat Alexa Fluor 594 (1:1000; Invitrogen, #A21209), donkey polyclonal anti-Rat Alexa Fluor 568 (1:1000; Abcam, Cat#175475), goat polyclonal anti-guinea Pig Alexa Fluor 488 (1:1000; Invitrogen, Cat#A110-73), goat polyclonal anti-guinea pig Alexa Fluor 647 (1:1000; Invitrogen, Cat#A21450), goat polyclonal anti-mouse HRP (1:10000; Jackson Immune Research Laboratories, PA, Cat#115-035-174), goat polyclonal anti-rat HRP (1:10,000; Jackson Immune Research Laboratories, #112-035-175), and mouse polyclonal anti-rabbit HRP (1:10,000; Jackson Immune Research Laboratories, #211-032-171).

The following conjugated antibodies were used: sheep polyclonal anti-digoxigenin alkaline phosphatase (1:2000-1:5000; Roche Life Science, Cat#11093274910), mouse monoclonal anti-βACTIN (clone AC-15) HRP (1:25000; Abcam, Cat#ab49900).

## Plasmids

Full-length *Susd4* mouse gene was cloned into the mammalian expression vector pEGFP-N1 (Addgene, MA, Cat#6085–1) to express a SUSD4-GFP fusion construct under the control of the CMV promoter (pSUSD4-GFP). An N-terminal HA tag was inserted just after the signal peptide (pHA-SUSD4-GFP). pHA-SUSD4 was obtained by removal of the C-terminal GFP of pHA-SUSD4-GFP. A truncated form of *Susd4*, expressing the HA-SUSD4-ΔC$_T$ mutant, was obtained by inserting a stop codon downstream of the sixth exon, 39 bp after the transmembrane domain using PCR on the pHA-SUSD4-GFP plasmid and the following primers: forward primer 5' GCG CTA GCG ATG TAT CCT TA T GAT GTT CCT G 3'; reverse primer 5'TAG CGG CCG CTA TTA GGG GGG GAA GTG GGC CTT 3'. Other mutant constructs were similarly obtained: the truncated form of HA-SUSD4-ΔN$_T$ corresponding to amino acids 294–490 and the extracellular form of SUSD4, HA-SUSD4-N$_T$, corresponding to amino acids 2–299. The HA-SUSD4-ΔPY contains a mutation in amino acids 411 and 414 changing PPAY to APAA while HA-SUSD4-ΔLY is mutated in amino acids 376 and 379 changing

LPTY to APTA. Mutagenesis was performed using the QuikChange Lightning Multi site directed mutagenesis kit (Agilent, Santa Clara, CA, Cat#210513) according to the manufacturer's instructions. The plasmid pIRES2-eGFP (Addgene, Cat#6029–1) was used as transfection control. The plasmid expressing SEPGluA2 (Addgene, Cat#24001) was used to follow GluA2. The control transmembrane protein PVRL3α was cloned into the mammalian expression vector pCAG-mGFP (Addgene, Cat#14757) to express the protein under the pCAG promoter (pCAG-PVRL3α).

## Viral mediated in vivo expression of HA-SUSD4

AAV2 particles were generated using an hSYN-DIO-HA-SUSD4-2A-eGFP-WPRE construct (Vector biolabs, Malvern, PA) and injected stereotaxically in cerebella of adult mice expressing the CRE recombinase in cerebellar PCs using the L7Cre mice. In the absence of Cre expression, the transgene is not produced. In the presence of Cre expression, the transgene will be 'FLip-EXchanged' leading to expression of the transgene specifically in PCs.

## In situ hybridization

Fresh frozen 20-µm-thick sections were prepared using a cryostat (Cryostar NX 70, Thermo Fisher Scientific, ref: 957000H) from brains of *Susd4* WT and KO mice at P0, P7, or P21. The probe sequence corresponded to the nucleotide residues 287–1064 bp (encompassing exons 2–5) for mouse *Susd4* (NM_144796.4) cDNA. The riboprobes were used at a final concentration of 0.05 µg/µL, and hybridization was done overnight at a temperature of 72°C. The anti-digoxigenin-AP antibody (for details, see 'Antibodies' section) was used at a dilution of 1:5000. Alkaline phosphatase detection was done using BCIP/NBT colorimetric revelation (Roche Life Science, Cat#11681451001).

## Behavioral study

Twelve- to 14-week-old male mice were used in this study. They were housed in groups of 3–5 in standard conditions: 12 hr light/dark cycle, with ad libitum food and water access. Seven days before the beginning of behavioral test, mice were housed individually to limit inter-houses variability resulting from social relationships. All behavioral tests took place in the light cycle.

### SHIRPA protocol

Mice performed a series of tests to ensure their general good health and motor performance and habituate them to being manipulated (*Crawley, 2006*). The test includes observation of appearance, spontaneous behavior, neurological reflexes, anxiety, motor coordination, balance rotarod, and muscular strength tests and were performed within 5 days. Individuals presenting deficits during the SHIRPA protocol were not used for other behavioral tests.

### Footprint analysis

The fore and hind paws of mice were dipped in magenta and cyan non-toxic paint, respectively. Mice were allowed to walk through a rectangular plastic tunnel (9 cm W × 57 cm L × 16 cm H), whose floor was covered with a sheet of white paper. Habituation was done the day before the test. Five footsteps were considered for the analysis. Footprints were scanned and length measurements were made using ImageJ.

### Rotarod

Mice were first habituated to the rotarod apparatus, 3 days before the acceleration test. The habituation protocol consists of 5 min at 4 r.p.m. To evaluate the motor coordination, mice were placed on immobile rotarod cylinders, which ramped up from 0 to 45 rotations per minute in 10 min. The timer was stopped when the mouse fell off the cylinder or did a whole turn with it. For a given session, this procedure was repeated three times per day separated by 60 min. The session was repeated during 5 consecutive days.

## Whole-cell patch-clamp on acute cerebellar slices

Responses to PF and CF stimulation were recorded in PCs of the lobule VI in acute parasagittal and horizontal (LTP experiments) cerebellar slices from *Susd4* KO juvenile (from P25 to P35) or adult (~P60) mice. *Susd4* WT littermates were used as controls. Mice were anesthetized using isoflurane

4% and sacrificed by decapitation. The cerebellum was dissected in ice-cold oxygenated (95% $O_2$ and 5% $CO_2$) bicarbonate buffered solution (BBS) containing (in mM): NaCl 120, KCl 3, $NaHCO_3$ 26, $NaH_2PO_4$ 1.25, $CaCl_2$ 2, $MgCl_2$ 1, and D(+)-glucose 35. Three-hundred-micrometer-thick cerebellar slices were cut with a vibratome (Microm HM650V: Thermo Scientific Microm, MA, or 7000smz-2 Campden Instruments Ltd., UK) in slicing solution (in mM): N-methyl-D-glucamine 93, KCl 2.5, $NaH_2PO_4$ 1.2, $NaHCO_3$ 30, HEPES 20, D(+)-glucose 25, $MgCl_2$ 10, sodium ascorbate 5, thiourea 2, sodium pyruvate 3, N-acetyl-cystein 1, kynurenic acid 1, and $CaCl_2$ 0.5 (pH 7.3). Immediately after cutting, slices were allowed to briefly recover at 37°C in the oxygenated sucrose-based buffer (in mM): sucrose 230, KCl 2.5, $NaHCO_3$ 26, $NaH_2PO_4$ 1.25, D(+)-glucose 25, $CaCl_2$ 0.8, and $MgCl_2$ 8. D-APV and minocycline at a final concentration of 50 µM and 50 nM, respectively, were added to the sucrose-based buffer. Slices were allowed to fully recover in bubbled BBS with 50 mM minocycline at 37°C for at least 40 min before starting the experiment, then maintained at room temperature for a maximum time of 8 hr (from slicing time). Patch-clamp borosilicate glass pipettes with 3–6 MΩ resistance were filled with the following internal solutions:

1. Cesium metanesulfonate solution (CsMe solution, for EPSC elicited from CF and PF), containing (in mM): $CsMeSO_3$ 135, NaCl 6, $MgCl_2$ 1, HEPES 10, MgATP 4, $Na_2GTP$ 0.4, EGTA 1.5, QX314Cl 5, TEA 5, and biocytin 2.6 (pH 7.3).
2. CsMe S-solution (for delayed EPSC quanta events), containing (in mM): $CsMeSO_3$ 140, $MgCl_2$ 0.5, HEPES 10, MgATP 4, $Na_2GTP$ 0.5, BAPTA 10, and neurobiotin 1% (pH 7.35).
3. Potassium gluconate solution (KGlu solution, for PF long-term plasticity), containing (in mM): K gluconate 136, KCl 10, HEPES 10, $MgCl_2$ 1, sucrose 16, MgATP 4, and $Na_2GTP$ 0.4 (pH 7.35).

Stimulation electrodes with ~5 MΩ resistances were pulled from borosilicate glass pipettes and filled with BBS. Recordings were performed at room temperature on slices continuously perfused with oxygenated BBS. The experiment started at least 20 min after the whole-cell configuration was established. The Digitimer DS3 (Digitimer Ltd) stimulator was used to elicit CF and PF and neuronal connectivity responses in PCs. Patch-clamp experiments were conducted in voltage clamp mode (except for the LTP and LTD induction protocols that were made under current-clamp mode) using a MultiClamp 700B amplifier (Molecular Devices, CA) and acquired using the freeware WinWCP written by John Dempster (https://pureportal.strath.ac.uk/en/datasets/strathclyde-electrophysiology-software-winwcp-winedr). Series resistance was compensated by 60–100% and cells were discarded if significant changes were detected. Currents were low-pass filtered at 2.2 kHz and digitized at 20 kHz.

## CF- and PF-EPSC experiments

To isolate the AMPARs current, the BBS was supplemented with (in mM): picrotoxin 0.1, D-AP5 10, CGP52432 0.001, JNJ16259685 0.002, DPCPX 0.0005, and AM251 0.001. CF and PF EPSCs were monitored at a holding potential of −10 mV. During CF recordings, the stimulation electrode was placed in the granule cell layer below the clamped cell; CF-mediated responses were identified by the typical all-or-none response and strong depression displayed by the second response elicited during paired-pulse stimulations (20 Hz). The number of CFs innervating the recorded PC was estimated from the number of discrete CF-EPSC steps. PF stimulation was achieved by placing the stimulation electrode in the molecular layer at the minimum distance required to avoid direct stimulation of the dendritic tree of the recorded PC. The input-output curve was obtained by incrementally increasing the stimulation strength. Peak EPSC values for PF were obtained following averaging of three consecutive recordings, values for CF EPSC correspond to the first recording. Short-term plasticity experiments were analyzed using a software written in Python by Antoine Valera (http://synaptiqs.wixsite.com/synaptiqs).

## PF long-term plasticity experiments

PCs were clamped at −60 mV. Each PF-induced response was monitored by a test protocol of paired stimulation pulses (20 Hz) applied every 20 s. A baseline was established during 10 min of paired-pulse stimulation in the voltage clamp configuration. After that, an induction protocol was applied in current-clamp mode with cells held at −60 mV. During LTD induction, the PFs were stimulated with two pulses at high frequency (200 Hz) and, after 100 ms, the CF was stimulated with four pulses at high frequency (200 Hz) repeated every 2 s for a period of 10 min. During LTP induction,

recordings were made using BBS not supplemented with picrotoxin, and the PFs were stimulated with bursts of 15 pulses at high frequency (100 Hz) repeated every 3 s for a period of 5 min (*Binda et al., 2016*). Then, PCs were switched to the voltage clamp mode and paired stimulation pulses applied again, lasting 40 min. All the data were normalized to the mean baseline. Long-term plasticity was analyzed with the software Igor Pro 6.05 (WaveMetrics INC, OR).

PF- and CF-delayed EPSC quanta events were detected and analyzed using the software Clampfit 10.7 (Molecular Devices). PF- and CF-delayed EPSC quanta superposed events were discarded manually based on the waveform. A threshold of 10 pA for minimal amplitude was used to select the CF events; 100 (PF) and 300 (CF) events for each neuron were studied by analyzing consecutive traces.

## High-density MEA analysis of PC spiking in acute cerebellar slices

Experiments were performed on acute cerebellar slices obtained from 3- to 6-month-old mice in artificial cerebrospinal fluid (ACSF) containing (in mM): NaCl 125, KCl 2.5, D(+)glucose 25, NaHCO$_3$ 25, NaH$_2$PO$_4$ 1.25, CaCl$_2$ 2, and MgCl$_2$ 1  (95% O$_2$ and 5% CO$_2$). Parasagittal slices (320 μm) were cut at 30°C (*Huang and Uusisaari, 2013*) with a vibratome (7000smz-2, Campden Instruments Ltd.) at an advance speed of 0.03 mm/s and vertical vibration set to 0.1–0.3 μm. Slices were then transferred to a chamber filled with oxygenated ACSF at 37°C and allowed to recover for 1 hr before recordings.

For recording, the slices were placed over a high-density microelectrode array of 4096 electrodes (electrode size, 21 × 21 μm; pitch, 42 μm; 64 × 64 matrix; Biocam X, 3Brain, Wädenswil, Switzerland) and constantly perfused with oxygenated ACSF at 37°C. Extracellular activity was digitized at 17 kHz and data were analyzed with the Brainwave software (3Brain). The signal was filtered with a Butterworth high-pass filter at 200 Hz, spikes were detected with a hard threshold set at −100 μV, and unsupervised spike sorting was done by the software. We selected units with a firing rate between 15 and 100 spikes per second and we excluded units presenting more than 5% of refractory period violation (set to 3 ms). Recordings were performed on two slices per animal, each slice containing between 20 and 200 active neurons, and results were then pooled for each animal.

To quantify the average variability in the firing rate, the CV of the ISI in seconds was calculated as the ratio of the standard deviation of ISIs to the mean ISI of a given cell. To measure the firing pattern variability within a short period of two ISIs, CV2 was calculated [CV2 = 2|ISI$_{n+1}$ − ISI$_n$|/(ISI$_{n+1}$ + ISI$_n$)] (*Holt and Douglas, 1996*).

## Affinity purification of SUSD4 interactors from synaptosome preparations

HEK293H (Gibco, MA, Cat#11631–017) were maintained at 37°C in a humidified incubator with 5% CO$_2$ in Dulbecco's modified Eagle's medium (DMEM; containing high glucose and glutamax, Life Technologies, Cat#31966047) supplemented with 10% fetal bovine serum (FBS, Gibco, Cat#16141–079) and 1% penicillin/streptomycin (Gibco, Cat#15140122). 10$^6$ cells were plated per well in a six-well plate and transfected 24 hr (h) after plating with the indicated plasmids (1 μg plasmid DNA per well) using Lipofectamine 2000 (Invitrogen, Cat#11668–019) according to manufacturer's instructions.

Forty-eight hours after transfection, cells were lysed and proteins were solubilized for 1 hr at 4°C under gentle rotation in lysis buffer (10 mM Tris-HCl pH 7.5, 10 mM EDTA, 150 mM NaCl, 1% Triton X100 [Tx; Sigma, Cat#x100], 0.1% SDS) supplemented with a protease inhibitor cocktail (1:100; Sigma, Cat#P8340) and MG132 (100 μM; Sigma, Cat#C2211). Lysates were sonicated for 10 s, further solubilized for 1 hr at 4°C and clarified by centrifugation at 6000 r.p.m. during 8 min (min). Supernatants were collected, incubated with 5 μg of rat monoclonal anti-HA antibody (for details, see 'Antibodies' section), together with 60 μL of protein G-sepharose beads (Sigma; Cat#10003D) for 3 hr at 4°C, to coat the beads with the HA-tagged SUSD4 proteins. When SUSD4-GFP was expressed for affinity pull downs, GFP-Trap was done according to the instructions of GFP-Trap_A (Chromotek, NY, Cat#ABIN509397). Coated beads were washed three times with 1 mL lysis buffer.

To prepare synaptosome fractions, cerebella from WT mice (P30) were homogenized at 4°C in 10 volumes (w/v) of 10 mM Tris buffer (pH 7.4) containing 0.32 M sucrose and protease inhibitor cocktail (1:100). The resulting homogenate was centrifuged at 800 g for 5 min at 4°C to remove nuclei and cellular debris. Synaptosomal fractions were purified by centrifugation for 20 min at 20,000 r. p.m. (SW41Ti rotor) at 4°C using Percoll-sucrose density gradients (2-6-10–20%; v/v). Each fraction

from the 10–20% interface was collected and washed in 10 mL of a 5 mM HEPES buffer pH 7.4 (NaOH) containing 140 mM NaCl, 3 mM KCl, 1.2 mM MgSO$_4$, 1.2 mM CaCl$_2$, 1 mM NaH$_2$PO$_4$, 5 mM NaHCO$_3$, and 10 mM D(+)-glucose by centrifugation. The suspension was immediately centrifuged at 10,000 g at 4°C for 10 min. Synaptosomes in the pellet were resuspended in 100 µL of lysis buffer (10 mM Tris-HCl pH 7.5, 10 mM EDTA, 150 mM NaCl, 1% Tx) supplemented with a protease inhibitor cocktail (1:100) and MG132 (100 µM). Lysates were sonicated for 10 s and further incubated for 1 hr at 4°C. HA-SUSD4, GFP-SUSD4, or its control GFP-coated beads were then incubated with the synaptosomal lysates for 3 hr at 4°C. Beads were washed three times with lysis buffer supplemented with 0.1% SDS. Bound proteins were eluted for 10 min at 75°C using Laemmli buffer (160 mM Tris pH 6.8, 4% SDS, 20% glycerol, 0.008% BBP) with 5% β-mercaptoethanol before SDS-PAGE followed by western blotting or mass spectrometry.

## Co-immunoprecipitation experiments in HEK293 cells

$10^6$ HEK293H cells were plated per well in six-well plates and transfected 24 hr after plating with the indicated plasmids (1.6 µg plasmid SEPGluA2 per well, using a molar ratio of 2:1 SEPGluA2:other plasmid) using Lipofectamine 2000 according to manufacturer's instructions. For anti-HA pull downs, proteins from HEK293 cell lysates were solubilized in lysis buffer (1 M Tris-HCl pH 8, 10 mM EDTA, 1.5 M NaCl, 1% Tergitol [Sigma; Cat#NP40], 2% Na azide, 10% SDS, and 10% Na deoxycholate) supplemented with a protease inhibitor cocktail (1:100) and MG132 (1%). Then, lysates were sonicated for 15 s, further clarified by a centrifugation at 14,000 r.p.m. for 10 min. Supernatants were collected and incubated with Dynabeads Protein G (Life Technologies, Cat#10004D) and 28.8 µg of rat monoclonal anti-HA antibody (for details, see 'Antibodies' section) under gentle rotation for 1 hr at 4°C. Precipitates were washed three times in lysis buffer and then eluted by boiling (65°C) the beads 15 min in sample buffer (made from sample buffer 2× concentrate, Sigma, Cat#S3401) before SDS-PAGE. For SEPGluA2 pull downs, 48 hr after transfection, cells were washed twice in 1× PBS, lysed with 200 µL of lysis buffer (50 mM Tris-HCl pH 8% and 1% Tx) supplemented with a protease inhibitor cocktail (1:100) and MG132 (50 µM), scraped, sonicated 3 × 5 s, and proteins were further solubilized for 30 min at 4°C under rotation. Lysates were clarified by centrifugation at 14,000 r.p.m. for 10 min at 4°C. Supernatants (inputs) were collected and incubated with G-protein Dynabeads (Thermo Fisher Scientific, Cat#10004D), previously linked to mouse anti-GFP antibody (for details, see 'Antibodies' section), under gentle rotation for 1 hr at 4°C, to coat the beads with the SEP-tagged GluA2 proteins and interactors. Using a magnet, coated beads were washed five times in lysis buffer and bound proteins were then eluted by boiling for 15 min at 65°C in 1× sample buffer before SDS-PAGE and western blot analysis for detection of HA-SUSD4, GluA2, and PVRL3α.

## Mass spectrometry analysis

Proteins were separated by SDS-PAGE on 10% polyacrylamide gels (Mini-PROTEAN TGX Precast Gels, Bio-Rad, Hercules, CA) and stained with Protein Staining Solution (Euromedex, Souffelweyer-sheim, France). Gel lanes were cut into five pieces and destained with 50 mM triethylammonium bicarbonate and three washes in 100% acetonitrile. Proteins were digested in-gel using trypsin (1.2 µg/band, Gold, Promega, Madison, WI), as previously described (*Thouvenot et al., 2008*). Digest products were dehydrated in a vacuum centrifuge.

### Nano-flow LC-MS/MS

Peptides, resuspended in 3 µL formic acid (0.1%, buffer A), were loaded onto a 15 cm reversed phase column (75 mm inner diameter, Acclaim Pepmap 100 C18, Thermo Fisher Scientific) and separated with an Ultimate 3000 RSLC system (Thermo Fisher Scientific) coupled to a Q Exactive Plus (Thermo Fisher Scientific) via a nano-electrospray source, using a 120 min gradient of 5-40% of buffer B (80% ACN, 0.1% formic acid) and a flow rate of 300 nL/min.

MS/MS analyses were performed in a data-dependent mode. Full scans (375–1500 m/z) were acquired in the Orbitrap mass analyzer with a 70,000 resolution at 200 m/z. For the full scans, 3 × $10^6$ ions were accumulated within a maximum injection time of 60 ms and detected in the Orbitrap analyzer. The 12 most intense ions with charge states ≥2 were sequentially isolated to a target value of 1 × $10^5$ with a maximum injection time of 45 ms and fragmented by higher-energy collisional

dissociation in the collision cell (normalized collision energy of 28%) and detected in the Orbitrap analyzer at 17,500 resolution.

## MS/MS data analysis

Raw spectra were processed using the MaxQuant environment (*Cox and Mann, 2008*, v1.5.5.1) and Andromeda for database search (*Cox et al., 2011*). The MS/MS spectra were matched against the UniProt Reference proteome (Proteome ID UP000000589) of *Mus musculus* (release 2017_03; http://www.uniprot.org) and 250 frequently observed contaminants (MaxQuant contaminants database) as well as reversed sequences of all entries. The following settings were applied for database interrogation: mass tolerance of 7 ppm (MS) and 0.5 Th (MS/MS), trypsin/P enzyme specificity, up to two missed cleavages allowed, only peptides with at least seven amino acids in length considered, and Oxidation (Met) and acetylation (protein N-term) as variable modifications. The 'match between runs' feature was allowed, with a matching time window of 0.7 min. False discovery rate (FDR) was set at 0.01 for peptides and proteins.

A representative protein ID in each protein group was automatically selected using an in-house bioinformatics tool (leading v2.1). First, proteins with the most numerous identified peptides are isolated in a 'match group' (proteins from the 'Protein IDs' column with the maximum number of 'peptides counts'). When more than one protein ID is present after filtering in a 'match group', the best annotated protein in UniProtKB (reviewed entries rather than automatic ones), highest evidence for protein existence, most annotated protein according to the number of Gene Ontology Annotations (GOA Mouse v151) is defined as the 'leading' protein. Only proteins identified with a minimum of two unique peptides, without MS/MS in control immunoprecipitation and exhibiting more than four-fold enrichment (assessed by spectral count ratio) in Sushi domain-containing protein 4 (SUSD4) immunoprecipitation vs. control immunoprecipitation, in the two biological replicates, were considered as potential partners of SUSD4 (*Table 1*).

## GO analysis

The statistically enriched GO categories for the 28 candidate proteins were determined by Cytoscape (v3.6) plugin ClueGO v2.5.3 (*Bindea et al., 2009*). The molecular function category was considered (release 18.12.2018, https://www.ebi.ac.uk/GOA), except evidences inferred from electronic annotations. Terms are selected by different filter criteria from the ontology source: three to eight GO level intervals, minimum of four genes per GO term, and 10% of associated genes/term. A two-sided hypergeometric test for enrichment analysis (Benjamini-Hochberg standard correction used for multiple testing) was applied against the whole identified protein as reference set. Other predefined settings were used. Each node representing a specific GO term is color-coded based on enrichment significance (p-value). Edge thickness represents the calculated score (κ) to determine the association strength between the terms.

## cLTD and GluA2 surface biotinylation assay in cerebellar acute slices

Three-hundred-micrometer-thick parasagittal cerebellar slices were obtained from P31-P69 WT and *Susd4* KO mice following the same protocol described before (see 'Whole-cell patch-clamp on acute cerebellar slices' section). Slices were incubated for 2 hr at 37°C in oxygenated BBS with or without proteasome (50 μM MG132 in DMSO) and lysosomal (100 μg/mL leupeptine in water, Sigma, Cat#11034626001) inhibitors. cLTD was induced by incubating the slices for 5 min at 37°C in BBS containing 50 mM K$^+$ and 10 μM glutamate (diluted in HCl), followed by a recovery period in BBS for 30 min at 37°C all under oxygenation, in presence or not of inhibitors. Control slices were incubated in parallel in BBS solution containing HCl. Slices were then homogenized in lysis buffer, containing: 50 mM Tris-HCl, 150 mM NaCl, 0.1% SDS, 0.02% Na azide, 0.5% Na deoxycholate, 1% NP-40, and protease inhibitor cocktail (1:100). Homogenates were incubated 45 min at 4°C, then sonicated and centrifuged at 14,000 r.p.m. for 10 min at 4°C. Supernatants were then heated at 65°C in 2× sample buffer (Sigma, Cat#S3401) prior to western blot analysis for detection of GluA2 and GluD2.

For GluA2 surface biotinylation assay, cerebellar slices (obtained from mice aged between P27 and P61) were treated as above. After a recovery period of 30 min at 37°C in BBS, slices were incubated in a biotinylation solution (Thermo Fisher Scientific, EZ-Link Sulfo-NHS-SS-Biotin,

Cat#A39258, 0.125 mg/mL) for 30 min on ice without oxygen. Slices were finally washed three times for 10 min in PBS pH 7.4 at 4°C and then homogenized in lysis buffer, containing: 50 mM Tris-HCl pH 8, 150 mM NaCl, 0.1% SDS, 0.02% Na azide, 0.5% Na deoxycholate, 1% NP-40, and protease inhibitor cocktail (1:100). Homogenates were incubated 45 min at 4°C, then sonicated and centrifuged at 14,000 r.p.m. for 10 min at 4°C. Supernatants (inputs) were collected and incubated with Dynabeads MyOne Streptavidin C1 (Thermo Fisher Scientific, Cat#65001) under gentle rotation overnight at 4°C. Using a magnet, beads were washed five times in lysis buffer and biotinylated proteins were then eluted by boiling for 15 min at 65°C in 1× sample buffer before SDS-PAGE and western blot analysis for detection of GluA2.

## Immunocytochemistry

Labeling of primary hippocampal neurons: Hippocampi were dissected from E18 mice embryos and dissociated. $1.2 \times 10^5$ neurons were plated onto 18 mm diameter glass coverslips precoated with 80 µg/mL poly-L-ornithine (Sigma, Cat#P3655) and maintained at 37°C in a 5% $CO_2$ humidified incubator in neurobasal medium (Gibco, Cat#21103049) supplemented with 2% B-27 supplement (Gibco, Cat#17504044) and 2 mM Glutamax (Gibco, Cat#35050-038). Fresh culture medium (neurobasal medium supplemented with 2% B-27, 2 mM L-glutamine [Gibco, Cat#A2916801], and 5% horse serum [Gibco, Cat#26050088]) was added every week for maintenance of the neuronal cultures.

Hippocampal neurons at days in vitro 13 (DIV13) were transfected using Lipofectamine 2000 and 0.5 µg plasmid DNA per well. After transfection, neurons were maintained in the incubator for 24 hr, then fixed with 100% methanol for 10 min at −20°C. After rinsing with PBS, non-specific binding sites were blocked using PBS containing 4% donkey serum (DS, Abcam, Cat#ab7475) and 0.2% Tx primary and secondary antibodies were diluted in PBS 1% DS/0.2% Tx and incubated 1 hr at room temperature. Three washes in PBS 0.2% Tx were performed before and after each antibody incubation. Nuclear counterstaining was performed with Hoechst 33342 (Sigma, Cat#14533) for 15 min at room temperature.

Labeling of primary cerebellar mixed cultures: Cerebellar mixed cultures were prepared from P0 tg/0 'B6.129-Tg(Pcp2-cre)2Mpin/J' (stock number 004146, outbred, C57Bl/6J background) mouse cerebella and were dissected and dissociated according to previously published protocol (*Tabata et al., 2000*). Neurons were seeded at a density of $5 \times 10^6$ cells/mL. Mixed cerebellar cultures were transduced at DIV3 using a Cre-dependent AAV construct that expresses HA-tagged SUSD4 and soluble GFP (2 µL of AAV2-hSYN-DIO-HA-SUSD4-2A-eGFP-WPRE at $4.1 \times 10^{12}$ GC/mL or control AAV2-hSYN-DIO-eGFP-WPRE at $5 \times 10^{12}$ GC/mL). At DIV17, neurons were fixed with 4% PFA in PBS 1× for 30 min at room temperature. After rinsing with PBS, non-specific binding sites were blocked using PBS containing 4% DS and 0.2% Tx. Primary and secondary antibodies were diluted in PBS 1% DS and 0.2% Tx and incubated 1 hr at room temperature. Three PBS 0.2% Tx washes were performed before and after each antibody incubation. Nuclear counterstaining was performed with Hoechst 33342 for 15 min at room temperature.

## Immunohistochemistry

### Labeling of brain sections

Thirty-micrometer-thick parasagittal brain sections were obtained using a freezing microtome (SM2010R, Leica) and brains obtained after intracardiac perfusion with 4% PFA in PBS solution of mice sedated with 100 mg/kg pentobarbital sodium. Sections were then washed three times for 5 min in PBS, then blocked with PBS 4% DS for 30 min. The primary antibodies were diluted in PBS, 1% DS, 1% Tx. The sections were incubated in the primary antibody solution overnight at 4°C and then washed three times for 5 min in PBS 1% Tx. Sections were incubated in the secondary antibody, diluted in PBS, 1% DS, 1% Tx solution for 1 hr at room temperature. The sections were then incubated for 15 min at room temperature with the nuclear marker Hoechst 33342 in PBS 0.2% Tx. Finally, the sections were washed three times for 5 min in PBS 1% Tx, recovered in PBS and mounted with Prolong Gold (Thermo Fisher Scientific, Cat#P36934) between microscope slides and coverslips (Menzel-gläser, Brunswick, Germany, Cat#15165252).

## RT-PCR and quantitative RT-PCR

For standard RT-PCR, total RNA was isolated from the cortex, cerebellum, and brainstem of 2-month-old *Susd4* KO mice and WT control littermates, using the RNeasy mini kit (Qiagen, Venlo, The Netherlands, Cat#74104). Equivalent amounts of total RNA (100 ng) were reverse-transcribed according to the protocol of SuperScript VILO cDNA Synthesis kit (Life Technologies, CA, Cat#11754–250) as stated by manufacturer's instructions. The primers used were forward 5' TGT TAC TGC TCG TCA TCC TGG 3' and reverse 5' GAG AGT CCC CTC TGC ACT TGG 3'. PCR was performed with an annealing temperature of 61°C, for 39 cycles, using the manufacturer's instructions (*Taq* polymerase; New England Biolabs, MA, Cat#M0273S). Quantitative PCR was performed using the TaqMan universal master mix II with UNG (Applied Biosystems, Cat# 4440038) and the following TaqMan probes: *Rpl13a* (#4331182_Mm01612986_gH) and *Susd4* (#4331182_Mm01312134_m1).

## Western blot analysis

After samples were mixed with sample buffer, proteins were resolved by electrophoresis on a 4–12% NuPAGE Bis-Tris-Gel according to Invitrogen protocols, then electrotransferred using TransBlot DS Semi-dry transfer Cell or TransBlot Turbo transfer system (Bio-Rad) to PVDF membrane (Immobilon-P transfer membrane, Millipore, Cat#IPVH00010). Membranes were blocked in PBS supplemented with Tween 0.2% (PBST) and non-fat milk 5% and incubated with primary antibodies in PBST-milk 5%. After washing three times in PBST, membranes were incubated with horseradish peroxidase-conjugated secondary antibodies in PBST-milk 5%. Membranes were finally washed three times and bound antibodies were revealed using Immobilon Western (Millipore, Cat#WBKLS) or Western Femto Maximum Sensitivity (Thermo Fisher Scientific, Cat#34095) or SuperSignal West Dura (Thermo Fisher Scientific, Cat#34075) or ECL western blotting substrate (Thermo Fisher Scientific, Cat#32209) chemiluminescent solutions and images acquired on a Fusion FX7 system (Vilber Lourmat, Île-de-France, France). Quantitation of western blots was performed using the ImageJ software on raw images under non-saturating conditions. Band intensities of proteins of interest were obtained after manually selecting a rectangular region around the band. The signal intensity of the band of interest was then normalized to the signal intensity of the corresponding βACTIN (used as a loading control). For quantifications of immunoprecipitation experiments, input intensities were normalized to βACTIN, and then the intensities of immunoprecipitated protein bands were normalized to the normalized inputs, unless otherwise stated.

## Image acquisition and quantification

In situ hybridization images were acquired using an Axio Zoom v16 (Zeiss, Oberkochen, Germany) microscope equipped with a digital camera (AxioCam HRm) using a 10× objective (pixel size 0.650 μm).

Immunofluorescence image stacks were acquired using a confocal microscope (SP5, Leica), using a 63× objective (1.4 NA, oil immersion, pixel size: 57 nm for cell culture imaging, pixel size: 228 nm for 63×; 76 nm, 57 nm, 45 nm for higher magnifications for in vivo imaging). The pinhole aperture was set to 1 airy unit and a z-step of 200 nm was used. Laser intensity and photomultiplier tube gain was set so as to occupy the full dynamic range of the detector. Images were acquired in 16-bit range. Immunofluorescence images and image stacks from *Figures 1C, D* and *4F* were acquired using a Zeiss LSM 980 Confocal with an Airyscan detector (v2.0), using a 63× objective (1.4 NA, oil immersion, pixel size: 43 nm, z-step of 150 nm).

Deconvolution was performed for the VGLUT1 images with Huygens 4.1 software (Scientific Volume Imaging) using maximum likelihood estimation algorithm from Matlab. Forty iterations were applied in classical mode, background intensity was averaged from the voxels with lowest intensity, and signal-to-noise ratio values were set to a value of 25.

VGLUT1 and VGLUT2 puncta were analyzed using the Matlab software and a homemade code source (Dr. Andréa Dumoulin). The number, area, and intensity of puncta were quantified using the mask of each puncta generated by the Multidimensional Image Analysis software from Metamorph (Molecular Devices). For each animal, puncta parameters were measured from four equidistant images within a 35-image stack at 160 nm interval, acquired from three different lobules (n = 12).

The software ImageJ was used to measure the total area of a cerebellar section from images of staining obtained with the nuclear marker Hoechst. The extension of the molecular layer was measured using images of the anti-CABP staining. Nine parasagittal sections were analyzed per animal. The data presented correspond to the mean per animal.

## Statistical analysis

Data from all experiments were imported in Prism (GraphPad Software, CA) for statistical analysis, except for electrophysiology data that were imported to Igor Pro 6.05 (WaveMetrics INC) for statistical analysis.

In the case of two column analyses of means, the differences between the two groups were assessed using two-tailed Student's t-test. Normality of populations was assessed using D'Agostino and Pearson, Shapiro-Wilk, and Kolmogorov-Smirnov normality tests. When groups did not fit the normal distribution, the non-parametric Mann-Whitney test was used. For the rotarod behavioral test (two variables, genotype and trial), two-way repeated measures ANOVA followed by Bonferroni post hoc test was performed. The two-tailed Student's one sample t-test (when normality criterion was met) or the two-tailed Wilcoxon signed rank test was used to compare ratios to a null hypothesis of 1 for biochemical experiments or 100 for long-term plasticity (*Fay, 2013*). Differences in cumulative probability were assessed with the Kolmogorov-Smirnov distribution test, and differences in distribution were tested using the chi-squared test.

## Acknowledgements

We gratefully acknowledge the Collège de France imaging facility (IMACHEM-IBiSA), in particular P Mailly for help with the design of the macro for GluA2 quantification and Estelle Anceaume for help with image acquisition. We also thank the personnel from the CIRB, INCI, and chronobiotron CNRS UMS 3415, IBPS, and IBENS animal facilities. We would like to thank Philippe Marin for advice on proteomics analysis. Mass spectrometry experiments were carried out using facilities of the Functional Proteomics Platform of Montpellier.

## Additional information

### Funding

| Funder | Grant reference number | Author |
|---|---|---|
| ATIP-AVENIR | RSE11005JSA | Fekrije Selimi |
| Idex PSL | ANR-10-IDEX-0001-02 PSL* | Fekrije Selimi |
| Agence Nationale de la Recherche | ANR 9139SAMA90010901 | Philippe Isope<br>Fekrije Selimi |
| Agence Nationale de la Recherche | ANR-15-CE37-0001-01 CeMod | Philippe Isope<br>Fekrije Selimi |
| Fondation pour la Recherche Médicale | DEQ20150331748 | Fekrije Selimi |
| Fondation pour la Recherche Médicale | DEQ20140329514 | Philippe Isope |
| H2020 European Research Council | SynID 724601 | Fekrije Selimi |
| Labex Memolife | ANR-10-LABX-54 MEMO LIFE | Keerthana Iyer |
| Ecole des Neurosciences de Paris | | Keerthana Iyer |
| Labex | ANR-11-IDEX-0004-02 | Laure Rondi-Reig |

The funders had no role in study design, data collection and interpretation, or the decision to submit the work for publication.

## Author contributions
Inés González-Calvo, Conceptualization, Formal analysis, Validation, Investigation, Visualization, Writing - original draft, Writing - review and editing; Keerthana Iyer, Conceptualization, Formal analysis, Validation, Investigation, Visualization, Writing - review and editing; Mélanie Carquin, Maxime Veleanu, Formal analysis, Validation, Investigation, Visualization, Writing - review and editing; Anouar Khayachi, Validation, Investigation, Writing - review and editing; Fernando A Giuliani, Sylvana Tahraoui, Mélanie Albert, Validation, Investigation; Séverine M Sigoillot, Validation, Investigation, Visualization, Methodology, Writing - review and editing; Jean Vincent, Formal analysis, Validation, Investigation; Martial Séveno, Formal analysis, Validation, Investigation, Methodology, Writing - review and editing; Oana Vigy, Validation, Investigation, Visualization; Célia Bosso-Lefèvre, Andréa Dumoulin, Investigation, Writing - review and editing; Yann Nadjar, Antoine Triller, Resources; Jean-Louis Bessereau, Resources, Writing - review and editing; Laure Rondi-Reig, Funding acquisition, Methodology, Writing - review and editing; Philippe Isope, Resources, Software, Supervision, Funding acquisition, Methodology, Writing - review and editing; Fekrije Selimi, Conceptualization, Resources, Software, Formal analysis, Supervision, Funding acquisition, Validation, Visualization, Methodology, Writing - original draft, Project administration, Writing - review and editing

## Author ORCIDs
Inés González-Calvo (ID) https://orcid.org/0000-0002-2652-2160
Keerthana Iyer (ID) https://orcid.org/0000-0002-4384-6781
Jean-Louis Bessereau (ID) https://orcid.org/0000-0002-3088-7621
Laure Rondi-Reig (ID) http://orcid.org/0000-0003-1006-0501
Philippe Isope (ID) http://orcid.org/0000-0002-0630-5935
Fekrije Selimi (ID) https://orcid.org/0000-0001-7704-5897

## Ethics
Animal experimentation: All animal protocols were approved by the Comité Regional d'Ethique en Experimentation Animale (no. 00057.01) and animals were housed in authorized facilities of the CIRB (# C75 05 12).

## Decision letter and Author response
Decision letter https://doi.org/10.7554/eLife.65712.sa1
Author response https://doi.org/10.7554/eLife.65712.sa2

# Additional files

## Supplementary files
• Supplementary file 1. Behavioral characterization of *Susd4* KO mice.
• Transparent reporting form

## Data availability
All data generated or analysed during this study are included in the manuscript and supporting files.

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
