## [Decision Letter]

**Acceptance summary:**

This is a very interesting paper that demonstrates the involvement of a specific protein degradation pathway in synaptic plasticity in the cerebellum. The authors show that SUSD4 is expressed throughout the brain and is abundant in cerebellar dendrites and spines. Mice with SUSD4 deletion have motor coordination and learning deficits, along with impaired LTD induction. This study provides novel insight into the uncharacterized role of SUSD4 and provides a detailed and well-performed analysis of the SUSD4 loss of function phenotype in the cerebellar circuit.

**Decision letter after peer review:**

Thank you for submitting your article "SUSD4 controls GLUA2 degradation, synaptic plasticity and motor learning" for consideration by *eLife*. Your article has been reviewed by three peer reviewers, and the evaluation has been overseen by Megan Carey as Reviewing Editor and Gary Westbrook as the Senior Editor. The reviewers have opted to remain anonymous.

Essential Revisions:

The paper is broad in scope, which was seen as both a strength and a weakness. In particular, the reviewers agreed that the cerebellar physiology was strong, but the proposed cellular/ molecular mechanism of degradation rather than internalization was intriguing, but not fully developed. There was also concern raised about the ease with which the authors move from a brain-wide SUSD4 expression pattern, to a focus exclusively on Purkinje cell physiology, and then to attempting to link that to behaviors that are cerebellum-dependent, but not cerebellum-specific. The Reviewers and Editors agreed that the paper could be suitable for *eLife* if the technical issues raised by the reviewers were addressed, as well as conclusions tempered regarding mechanism and cerebellar specificity/ behavior.

1) Address the points of reviewer 1 regarding electrophysiological experiments.

2) The proposed mechanism is interesting, but preliminary. The authors should tone down significantly their mechanistic conclusions, and present the data factually, without trying to make a bigger "story" out of them than it is possible at this point.

3) In addition, please address the technical points raised regarding mechanism, particularly those of reviewer 3 regarding the IP in Figure 5.

4) The authors have not demonstrated clear links between the physiological results and the behavioral deficits. Especially within the context of brain-wide expression patterns for SUSD4, the authors should directly address this limitation and avoid implied causality.

5) *eLife* policy for titles states: Authors should avoid acronyms in the title:

Titles of *eLife* research papers should avoid unfamiliar abbreviations or acronyms, or authors should spell out in full or provide a brief explanation for any acronyms. Please revise your title with this advice in mind.

Reviewer #2 (Recommendations for the authors):

Figure 3A-C can be moved to supplementary, it's negative findings that are not directly pertaining to the conclusions.

The authors often use term like "could", "might" which casts a bit of uncertainty on their confidence on their conclusions. It would be better to either provide data that either support or dismiss a model, and state the conclusions in that manner.

GLUA2 is not the correct terminology, is should be GluA2

---

## [Author Response]

Essential Revisions:The paper is broad in scope, which was seen as both a strength and a weakness. In particular, the reviewers agreed that the cerebellar physiology was strong, but the proposed cellular/ molecular mechanism of degradation rather than internalization was intriguing, but not fully developed. There was also concern raised about the ease with which the authors move from a brain-wide SUSD4 expression pattern, to a focus exclusively on Purkinje cell physiology, and then to attempting to link that to behaviors that are cerebellum-dependent, but not cerebellum-specific. The Reviewers and Editors agreed that the paper could be suitable for eLife if the technical issues raised by the reviewers were addressed, as well as conclusions tempered regarding mechanism and cerebellar specificity/ behavior.1) Address the points of reviewer 1 regarding electrophysiological experiments.

We have addressed the points of reviewer 1 regarding electrophysiological experiments: 1) by modifying the text in the Results section regarding the LTP induction; 2) By adding an analysis of the complex spike waveform during LTD induction in current clamp mode (Figure 3—figure supplement 1).

2) The proposed mechanism is interesting, but preliminary. The authors should tone down significantly their mechanistic conclusions, and present the data factually, without trying to make a bigger "story" out of them than it is possible at this point.

We have changed the text to address this concern, in particular the end of the Abstract, Introduction, the Results section and the Discussion.

3) In addition, please address the technical points raised regarding mechanism, particularly those of reviewer 3 regarding the IP in Figure 5.

We have modified Figure 5D to show the amount of co-immunoprecipitated GluA2 normalized to the input GluA2 and relative to the amount of immunoprecipitated HA-tagged SUSD4 construct. We have also added the quantifications of the inputs in Figure 5—figure supplement 2.

We have added a new experiment (Figure 4—figure supplement 2) that provides the control requested by reviewer 3 for non-specific binding of overexpressed membrane proteins. We have performed the reverse co-immunoprecipitation experiment using SEP-GluA2 as a bait and probed for HA-SUSD4 or control PVRL3α in the affinity-purified extracts. While both HA-SUSD4 and PVRL3α are readily detected in the inputs, only HA-SUSD4 is detected in the immunopurified extracts confirming the specific interaction between SEP-GluA2 and HA-SUSD4 in transfected HEK293 cells.

4) The authors have not demonstrated clear links between the physiological results and the behavioral deficits. Especially within the context of brain-wide expression patterns for SUSD4, the authors should directly address this limitation and avoid implied causality.

We are sorry if we have unintentionally conveyed the idea that the behavioral phenotype is a direct result of synaptic deficits in Purkinje cells. This can indeed only be demonstrated using a conditional invalidation of *Susd4* in Purkinje cells. We have modified the text accordingly to avoid implied causality. We have also directly addressed this limitation in the Discussion.

5) eLife policy for titles states: Authors should avoid acronyms in the title:Titles of eLife research papers should avoid unfamiliar abbreviations or acronyms, or authors should spell out in full or provide a brief explanation for any acronyms. Please revise your title with this advice in mind.

We have modified the title to remove abbreviations.

Reviewer #2 (Recommendations for the authors):Figure 3A-C can be moved to supplementary, it's negative findings that are not directly pertaining to the conclusions.

We have moved Figure 3A and 3B to the supplementary. We have kept Figure 3C in the main, since while the results are negative, we think that they are important to properly interpret the electrophysiology data.

The authors often use term like "could", "might" which casts a bit of uncertainty on their confidence on their conclusions. It would be better to either provide data that either support or dismiss a model, and state the conclusions in that manner.

We have changed the text accordingly.

GLUA2 is not the correct terminology, is should be GluA2

As requested, we have corrected the text and are using GluA2 and GluD2 when referring to glutamate receptor subunits. For the rest of the genes/proteins referred to in this study, we have kept the format defined by the International Committee on Standardized Genetic Nomenclature for Mice (http://www.informatics.jax.org/mgihome/nomen/gene.shtml#gene_sym): “Protein symbols use all uppercase letters.”